# Linear Convergence of SGD on Overparametrized Shallow Neural Networks

## Abstract

Despite the non-convex landscape, first-order methods can be shown to reach global minima when training overparameterized neural networks, where the number of parameters far exceed the number of training data. In this work, we prove linear convergence of stochastic gradient descent when training a two-layer neural network with smooth activations. While the existing theory either requires a high degree of overparameterization or non-standard initialization and training strategies, e.g., training only a single layer, we show that a subquadratic scaling on the width is sufficient under standard initialization and training both layers simultaneously if the minibatch size is sufficiently large and it also grows with the number of training examples. Via the batch size, our results interpolate between the state-of-the-art subquadratic results for gradient descent and the quadratic results in the worst case.

## 1 Introduction

Our understanding of the optimization landscape of supervised learning with neural networks has vastly improved in recent years. This is in part due to the observation that *overparameterization* is key to overcome the pitfalls of first-order methods in general non-convex problems (Soltanolkotabi et al., 2019). Under this assumption, a line of research has established convergence of first-order methods such as gradient descent (GD) to global optimality, (Allen-Zhu et al., 2019; Kawaguchi & Huang, 2019; Du et al., 2019; Du & Lee, 2018; Zou & Gu, 2019; Brutzkus & Globerson, 2017; Song & Yang, 2019; Oymak & Soltanolkotabi, 2020), a phenomenon that has been confirmed in practice.

Empirically, as long as the width of a network scales linearly with the size of the training data (mild overparameterization), stochastic gradient descent (SGD) enjoys fast convergence to global optimality (Livni et al., 2014; Safran & Shamir, 2018; Oymak & Soltanolkotabi, 2020; Kawaguchi & Huang, 2019). *Can we explain such behavior theoretically?* Sadly, the available characterizations require a larger degree of overparameterization, or imposes additional assumptions, which do not hold for the algorithms that are used in practice. For example, if GD is applied exclusively to the last layer, Kawaguchi & Huang (2019) show that an ideal linear scaling of the width is sufficient to guarantee convergence. Song & Yang (2019) prove quadratic scaling when GD is applied only to the first layer.

For two-layer neural networks, when both layers are trained with GD simultaneously, state-of-the-art results show that subquadratic (not linear) scaling is enough to converge to global optimality (Anonymous). Despite being close to the ideal linear rate of overparameterization, due to computational constraints, GD is rarely used in modern applications involving huge datasets. Hence, closing the gap between theory and practice requires studying scalable first-order algorithms such as SGD. Our work focuses on mini-batch SGD, which is one of the most common algorithms for training deep models. We study convergence of SGD when it is applied to train both layers of a neural network, which is initialized with standard initialization schemes.

**Our contributions:**

- We show that under proper initialization and choice of learning rate, the iterates of SGD converge to a global minimum with high probability and exponentially fast for a general non-convex problem assuming that the loss function satisfies a growth condition.
- For the special case a two-layer neural network, we show that a subquadratic scaling on the width is sufficient under standard initialization and training both layers simultaneously, if the

| Reference | Algorithm | Activation | Setting | Scaling |
|-----------|-----------|------------|---------|---------|
| Oymak & Soltanolkotabi (2020) | SGD on layer 1 | ReLU | QL | $\tilde{\Omega}(n^2)$ |
| Song & Yang (2019) | GD on layer 1 | ReLU | SD | $\tilde{\Omega}(n^2)$ |
| Kawaguchi & Huang (2019) | GD on layer 2 | ReLU | CLL | $\tilde{\Omega}(n)$ |
| Du et al. (2019) | GD | ReLU | SD+QL | $\tilde{\Omega}(n^6)$ |
| Zou & Gu (2019) | GD | ReLU | SD+QL | $\Omega(n^8)$ |
| Anonymous | GD | smooth | QL | $\tilde{\Omega}(n^{\frac{3}{2}})$ or $\tilde{\Omega}(n^2)$ |
| Allen-Zhu et al. (2019) | SGD | ReLU | SD+QL | $\Omega(n^{24})$ |
| **This paper** | SGD | smooth | QL | $\tilde{\Omega}(n^{\frac{3}{2}})$ or $\tilde{\Omega}(n^2)$ |

**Table 1:** Required degree of Overparameterization for training shallow networks with global convergence guarantees. QL=quadratic loss, CLL=convex and Lipschitz loss, SD=separable data. The notation $\tilde{\Omega}$ ignores logarithmic factors.

minibatch size is sufficiently large and it also grows with the number of training examples. For constant batch size, we show that quadratic overparametrization is sufficient. Our results interpolate between subquadratic and quadratic scalings depending on the batch size.

**Related work.** The majority of the existing literature on overparameterization focuses on GD (Du et al., 2019; Du & Lee, 2018; Allen-Zhu et al., 2019; Zou & Gu, 2019). Allen-Zhu et al. (2019) provided theoretical bounds for deep networks trained with SGD. However, their results require an overparameterization degree that is too large, compared to what can be achieved for GD. In contrast, we study SGD and how the batch size affects the required degree of overparameterization. Chen et al. (2021) establish generalization guarantees and sufficient network width when SGD trains deep ReLU networks for binary classification, which is a different setting compared to our paper.

We study the case where SGD updates *all* the parameters of a shallow neural network. In contrast, a number of existing literature assume that only the parameters corresponding to some layers are updated throughout training (Oymak & Soltanolkotabi, 2020; Kawaguchi & Huang, 2019; Song & Yang, 2019). When SGD is applied only to the first layer, Oymak & Soltanolkotabi (2020) showed that quadratic scaling is sufficient for convergence with linear rate. Despite being an interesting theoretical setup, such algorithmic choice rarely happens in practice.

There are also differences regarding the choice of activation function. While ReLU can be considered as the default activation function when studying deep neural networks, its non-smoothness may be the reason why results for ReLU networks require substantially more number of parameters or additional assumptions on the data (like separability) to guarantee convergence to a global minimum. Moreover, backpropagation on ReLU networks does not correctly calculate the gradient at all points of differentiability (Kakade & Lee, 2018; Bolte & Pauwels, 2021), which raises major technical issues. In contrast, we assume a smooth activation, similar to Anonymous, which avoids such issues and achieves lower overparameterization degrees.

The authors of (Anonymous) established subquadratic scaling when GD trains a shallow neural network. In this paper, we focus on SGD, which results in substantial technical challenges. Compared to the results in (Anonymous), controlling the length of the trajectory is more involved in this paper, which requires a new analysis technique that bounds the length of the trajectory with high probability. We consider the effect of mini-batch SGD, which shows an interpolation between subquadratic and quadratic scaling. We also improve the estimates at initialization and show that more relaxed assumptions are sufficient to establish sufficient overparameterization degree.

We summarize such recent results in the *overparametrization* literature in Table 1.

**Lazy Training.** Proving fast convergence to global optimality is not a complete answer. It has been shown that despite fast convergence, it is possible that an algorithm tends towards a solution with poor performance on test data, if the training falls in the so-called *Lazy Training* regime (Chizat et al., 2019). Thus, any useful algorithmic framework for learning neural networks should avoid this regime, usually through careful initialization schemes. For example, despite requiring only linear overparametrization for GD, the initialization studied by Nguyen & Mondelli (2020) leads to

the lazy regime. This is the reason why we omit such result from Table 1. In this work, we show global convergence and achieve subquadratic scaling under standard initialization schemes, which empirically perform well on test data.

Polylogarithmic width is enough to obtain convergence for neural networks of arbitrary depth, according to Ji & Telgarsky (2020); Chen et al. (2021). However, in those work, convergence is understood in an ergodic sense. This is a weaker notion than strict convergence with high probability, which is the one we consider, and which better matches practical applications.

Given perfect knowledge about the underlying function that generates the labels and under the assumption that such target function has low-rank approximation, Su & Yang (2019) showed that GD achieves zero-approximation. This is different from the problem considered in our paper.

For a binary classification problem, Daniely (2020) showed that near linear network size is sufficient for SGD to memorize random examples under a variant of Xavier initialization, which is a different setting compared to our paper. For a deep neural network with pyramidal structure and smooth activations, Nguyen & Mondelli (2020) showed that subquadratic scaling is sufficient for global convergence of GD under a restrictive initialization scheme. In this paper, we establish global convergence for SGD under standard initialization.

A recent line of work uses mean-field analysis to approximate the target distribution of the weights in a neural network via their empirical distribution (Mei et al., 2019; Lu et al., 2020). Nevertheless, such results do not provide useful overparametrization degree bounds in terms of the number of samples. In contrast, our work does not require such approximations and we focus on deriving explicit sufficient overparametrization rates for global optimality of SGD.

**Notation.** We use $\| \cdot \|$ to denote the Euclidean norm of a vector and Frobenius norm of a matrix. We use $\nabla$ to represent the Jacobian of a vector-valued and gradient of a scalar-valued function. We use $\odot$ and $\otimes$ to represent the entry-wise Hadamard product and Kronecker product, respectively. We use lower-case bold font to denote vectors. We use calligraphic and standard fonts to represent sets and scalars, respectively. We use $\sigma_{\min}(T)$ and $\sigma_{\max}(T)$ to denote the smallest and largest singular values of a linear map $T$. We use $[n]$ to represent $\{1, \cdots, n\}$ for an integer $n$. We use $\tilde{O}$ and $\tilde{\Omega}$ to hide logarithmic factors and use $\lesssim$ to ignore terms up to constant and logarithmic factors.

## 2 A GENERAL GLOBAL CONVERGENCE RESULT FOR SGD

In this section, we consider a general non-convex minimization problem and show that for a certain choice of learning rate and careful initialization, the iterates of SGD converge to a global minimum with high probability and exponentially fast. In Section 3, we extend our consideration to the training of a shallow neural network and find the hidden layer size, which is sufficient for SGD to converge to a global minimum *i.e.,* its overparameterization degree.

**Definition 1** (Smoothness). *Let $\beta_\psi > 0$. A function $\psi : \mathbb{R}^{d_1} \to \mathbb{R}^{d_2}$ is $\beta_\psi$-smooth, if for all $\mathbf{u}, \mathbf{v} \in \mathbb{R}^{d_1}$, we have*

$$\sigma_{\max}(\nabla\psi(\mathbf{u}) - \nabla\psi(\mathbf{v})) \leq \beta_\psi \|\mathbf{u} - \mathbf{v}\|. \tag{1}$$

**Definition 2** (PL condition (Bolte et al., 2017)). *A function $\psi : \mathbb{R}^{d_1} \to \mathbb{R}$ satisfies the PL condition if there exists $\alpha_\psi > 0$ such that, for all $\mathbf{u} \in \mathbb{R}^{d_1}$, we have*

$$\psi(\mathbf{u}) \leq \frac{\|\nabla\psi(\mathbf{u})\|^2}{2\alpha_\psi}. \tag{2}$$

We are now ready to state our finite-sum compositional optimization problem:

$$\min_{\mathbf{w}\in\mathbb{R}^d} \left\{ h(\mathbf{w}) := f(\Phi(\mathbf{w})) = \frac{1}{m}\sum_{j=1}^{m} f_j(\Phi(\mathbf{w})) \right\}, \qquad \Phi : \mathbb{R}^d \to \mathbb{R}^{\tilde{d}}, \qquad f, f_j : \mathbb{R}^{\tilde{d}} \to \mathbb{R}_+ \tag{3}$$

where $m$ denotes the number of training examples.

**Assumption 1.** *The functions introduced in Eq. (3) satisfy the following properties: (i) $\Phi$ is twice-differentiable and $\beta_\Phi$-smooth (Definition 1), (ii) $f$ is twice-differentiable and $\beta_f$-smooth and (iii) $f$ satisfies the PL condition with some $\alpha_f > 0$ (Definition 2).*

We study the iterates of the stochastic gradient descent (SGD) algorithm when applied to the objective function $h$ in Eq. (3). For $i \geq 0$, let $\mathcal{I}^i$ denote a random minibatch at iteration $i$ drawn uniformly at random, independent of all previous draws. Let $b \in [m]$ denote the minibatch size, *i.e.,* $|\mathcal{I}^i| = b$ for all $i$. The SGD iterates are defined by a random variable $\mathbf{w}^0$, referred to as the *initialization*, and the update rule:

$$\mathbf{w}^{i+1} = \mathbf{w}^i - \lambda \frac{1}{b} \sum_{j \in \mathcal{I}^i} \nabla h_j(\mathbf{w}^i), \tag{4}$$

where $\lambda > 0$ is the learning rate and $h_j(\mathbf{w}) \coloneqq f_j(\Phi(\mathbf{w})) \ \forall j, \mathbf{w}$.

An important feature of the SGD iterates is that $\frac{1}{b} \sum_{j \in \mathcal{I}^i} \nabla h_j(\Phi(\mathbf{w}^i))$ in Eq. (4) is an unbiased estimator of the gradient $\nabla h(\mathbf{w}^i)$ given $\mathbf{w}^i$, *i.e.,* $\mathbb{E}\left[\frac{1}{b} \sum_{j \in \mathcal{I}^i} \nabla h_j(\Phi(\mathbf{w}^i)) | \mathbf{w}^i\right] = \nabla h(\mathbf{w}^i)$. Nevertheless, this is not enough for SGD to converge to the first-order optimality. In addition, we will assume that $f$ in Eq. (3) satisfies the growth condition (Schmidt & Roux, 2013; Vaswani et al., 2019; Cevher & Vu, 2019):

**Definition 3** (Growth condition). *A function $\psi : \mathbb{R}^d \to \mathbb{R}$ with a finite-sum structure satisfies the growth condition with minibatch size $b$ if there exists $\eta_\psi > 0$ such that, for all $\mathbf{u} \in \mathbb{R}^d$, we have*

$$\mathbb{E}\left[\left\|\frac{1}{c} \sum_{j \in \mathcal{I}} \nabla \psi_j(\mathbf{u})\right\|^2\right] \leq \eta_\psi \|\nabla \psi(\mathbf{u})\|^2, \tag{5}$$

*where the expectation is over the random choice of set $\mathcal{I}$.*

**Assumption 2.** *In Eq. (3), $f$ satisfies the growth condition (Definition 3) for some $\eta_f > 0$.*

We are now ready to state the main result of this section.

**Theorem 1.** *Let Assumptions 1 and 2 hold and let $\zeta > 1$. Suppose that at initialization,*

$$0 < \mu_\Phi \leq \sigma_{\min}(\nabla \Phi^*(\mathbf{w}^0)) \leq \sigma_{\max}(\nabla \Phi^*(\mathbf{w}^0)) \leq \nu_\Phi, \quad h(\mathbf{w}^0) = O\left(\frac{\alpha_f \mu_\Phi^6}{\zeta \beta_\Phi^2 \eta_f \nu_\Phi^2}\right). \tag{6}$$

*Then, for a sufficiently small learning rate*

$$\lambda \lesssim \min\left(\frac{\mu_\Phi^2}{\eta_f(\beta_\Phi \nu_\Phi^2 \|\nabla f(\Phi(\mathbf{w}^0))\| + \beta_f \nu_\Phi^4 + \beta_f \mu_\Phi \nu_\Phi^3)}, \frac{\mu_\Phi}{\zeta \sqrt{\eta_f} \nu_\Phi (\beta_\Phi \|\nabla f(\Phi(\mathbf{w}^0))\| + \beta_f \nu_\Phi \mu_\Phi)}\right) \tag{7}$$

*the iterates of SGD $\{\mathbf{w}^i\}_{i \geq 0}$ (4) converge to a global minimizer of $h$ (3) with the optimal value of zero, exponentially fast and with probability at least $1 - 1/\zeta$. The rate of convergence is given by*

$$\mathbb{E}[h(\mathbf{w}^i)] \leq (1 - C\lambda \alpha_f \mu_\Phi^2)^i \cdot h(\mathbf{w}^0)$$

*for a universal constant $C$.*

**Remark 1.** *The second item in Eq. (6) suggests initializing close to a global minimum of the non-convex optimization problem. This feature has precedence in the related literature, e.g., in matrix factorization (Chi et al., 2019).*

The proof of Theorem 1 is deferred to Appendix B. However, in the remaining of this section we provide a sketch of the main arguments that lead to the result. The first condition in Eq. (6) is central to our arguments, and we will refer to it as the *near-isometry* property.

**Definition 4** (Near-isometry). *A linear map $T : \mathbb{R}^{d_1} \to \mathbb{R}^{d_2}$ is $(\mu, \nu)$-near-isometry if there exist $0 < \mu \leq \nu$ such that*

$$\mu \leq \sigma_{\min}(T) \leq \sigma_{\max}(T) \leq \nu. \tag{8}$$

Let $\overline{\mathbf{w}}$ denote the limit point when the SGD algorithm is run with some learning rate and let $\nabla \Phi^*(\overline{\mathbf{w}})$ denote the adjoint operator of $\nabla \Phi(\overline{\mathbf{w}})$. Convergence of SGD is ensured with high probability due to the strong growth condition (Definition 3) along with proper learning rate and initialization. We note that $\overline{\mathbf{w}}$ is a first-order stationary point of $h$. Hence we have:

$$0 = \nabla h(\overline{\mathbf{w}}) = \nabla \Phi^*(\overline{\mathbf{w}}) \nabla f(\Phi(\overline{\mathbf{w}})) \tag{9}$$

Note that if $\nabla\Phi^*(\overline{\mathbf{w}})$ is nonsingular, it would follow that $\nabla f(\Phi(\overline{\mathbf{w}})) = 0$. The PL condition (Definition 2) would then imply that $\Phi(\overline{\mathbf{w}})$ is a global minimizer of $f$ and hence, a global minimizer of $h$. With this fact in mind, our proof can be summarized in three steps: first, a careful choice of initialization will ensure that $\nabla\Phi^*$ is nonsingular for all elements within a certain distance of $\mathbf{w}^0$. Second, we show that under small enough learning rate, the iterates of SGD remain close to the initialization $\mathbf{w}^0$, with high probability regardless of the number of iterations.

The third and final step will use the non-singularity of $\nabla\Phi^*$ at convergence and Eq. (9) to conclude global optimality.

This is akin to the arguments in (Anonymous), however, in our case the stochasticity in the SGD updates poses a challenge for controlling the distance to initialization. We use concentration bounds on the length of the path and show that the SGD trajectory remains in the region where $\nabla\Phi^*$ is non-singular, with high probability.

A crucial result for the first step of our proof has already been established in by (Anonymous). It shows that a smooth function that is near-isometry at initialization will preserve such property for all points within a certain distance.

**Lemma 1** (Anonymous). *Let $\Phi$ be $\beta_\Phi$-smooth and $\nabla\Phi^*(\mathbf{w}_0)$ be a $(\mu_\Phi, \nu_\Phi)$-near-isometry. Then*

$$\text{for all } \mathbf{w} \text{ such that } \|\mathbf{w} - \mathbf{w}_0\| \leq \frac{\mu_\Phi}{2\beta_\Phi}, \quad \frac{\mu_\Phi}{2} \leq \sigma_{\min}(\nabla\Phi^*(\mathbf{w})) \leq \sigma_{\max}(\nabla\Phi^*(\mathbf{w})) \leq \frac{3\nu_\Phi}{2} \quad (10)$$

The second step in the proof of Theorem 1 is to compute the expected length of the SGD trajectory which is spent inside the ball defined in Eq. (10). We find an upper bound on this expected length depending on the initialization and learning rate, but independent of the number of iterations. Hence, under some proper initialization and learning rate, we can control the expected length of the trajectory for which Lemma 1 holds. In particular, we have

**Proposition 1** (Expected length of trajectory). *Let Assumptions 1 and 2 hold and let $\zeta > 1$. Let the random variable $I$ denote the first iteration of SGD (Eq. (4)) such that*

$$\mathbf{w}^I \notin B := \mathrm{ball}(\mathbf{w}^0, \rho_\Phi) := \{\mathbf{w} : \|\mathbf{w} - \mathbf{w}^0\| \leq \rho_\Phi\} \quad (11)$$

*or $I = \infty$ if the trajectory does not leave $B$. Suppose that $\mathbf{w}^0$ satisfies Eq. (6) and SGD is executed with sufficiently small learning rate, which satisfies Eq. (7). An upper bound on the expected length of the SGD trajectory is given by*

$$\mathbb{E}[\ell(I)] \leq \frac{\mu_\Phi}{2\zeta\beta_\Phi} = \frac{\rho_\Phi}{\zeta}. \quad (12)$$

We provide the sketch of the proof (see Appendix A for the complete proof). We first find an upper bound on the expected length of the trajectory in terms of the norm of gradients of $f$. With a proper learning rate, we find an upper bound on the norm of the gradient in terms of the expected decent of $f$ in two consecutive iterates, which are inside the ball. We also ensure that the learning rate is sufficiently small such that $\mathbb{E}[\|\mathbf{w}^I - \mathbf{w}^{I-1}\|]$ is bounded. Finally, under proper initialization, we obtain an upper bound on the expected length of the trajectory for the iterates inside the ball, *i.e.,* $\mathbb{E}[\sum_{i=0}^{I-1} \|\mathbf{w}^i - \mathbf{w}^{i-1}\|]$.

**Remark 2.** *A similar phenomenon that shows bounded length of the trajectory has been observed in various settings mainly for gradient descent (Du et al., 2019; Oymak & Soltanolkotabi, 2019; Anonymous). In this paper, we focus on a compositional non-convex problem trained with SGD, which is more challenging to analyze.*

Using the upper bound (12) on the expected length of the trajectory spent inside $B = \mathrm{ball}(\mathbf{w}^0, \rho_\Phi)$, we can bound the probability that the SGD iterates leaves the ball $B$. Indeed, in order for the process to leave $B$ starting from $\mathbf{w}^0$, it is required that the length of the trajectory spent inside $B$ satisfies $l(I) \geq \rho_\Phi$. Hence, using bound (12) on $\mathbb{E}[\ell(I)]$ together with a concentration bound (Markov inequality in our case), we can upper bound the probability of SGD iterates leaving $B$. Finally, under the event that the SGD iterates remain in $B$, an upper bound on $\mathbb{E}[\ell(I)]$ implies the convergence of the iterates.

**Remark 3.** *With a more involved analysis on the concentration properties of the random variable $\ell(I)$, it may be possible to greatly improve the dependence of the initialization and step size on $\zeta$.*

*Indeed, the current analysis assumes the worst-case scenario, where the SGD iterates either remain at the initialization, or directly leave the ball $B$ in a straight line (this scenario indeed maximizes the probability that the process leaves the ball, given a bound on $\mathbb{E}[\ell(I)]$).*

*Although $\ell(I)$ is obtained as a sum of random variables, the difficulty of obtaining better concentration bounds for $\ell(I)$ comes from the high level of dependence between all the variables involved. A better analysis would thus need to better understand how the trajectories behave inside the ball $B$, e.g., by bounding the variance of $\ell(I)$.*

In the following section, we specify our result to the special case of shallow neural networks. We will show that, in the case of quadratic loss, the strong growth condition naturally holds, with a constant depending on the batch size. Moreover, using Gaussian initialization for the neural network parameters, we can control the initial smoothness and near-isometry parameters involved in Theorem 1 with high probability.

## 3 GLOBAL OPTIMALITY OF NEURAL NETWORKS TRAINED WITH SGD

**Setup.** We will consider the problem of training a shallow neural network with one hidden layer, input dimension $d_0$, $d_1$ hidden nodes, output dimension $d_2$, and quadratic loss. We denote the data and label matrices as $X \in \mathbb{R}^{d_0 \times m}$ and $Y \in \mathbb{R}^{d_2 \times m}$, respectively.

Let $W \in \mathbb{R}^{d_1 \times d_0}$ and $V \in \mathbb{R}^{d_2 \times d_1}$ denote the parameters of the first and second layers of the network, respectively. We collect both parameters in a variable $\Theta = (W, V) \in \mathbb{R}^{d_1 \times d_0} \times \mathbb{R}^{d_2 \times d_1}$. In order to fit the supervised training of the network to the template studied in Section 2 (Eq. (3)) we define:

$$\Phi(\Theta) := V \cdot \phi(WX) \in \mathbb{R}^{d_2 \times m}, \qquad f_j(Z) = \|Z_j - Y_j\|^2 \in \mathbb{R}_+ \tag{13}$$

where $Z_j$ denotes the $j$-th column of a matrix $Z$ and $\phi : \mathbb{R} \to \mathbb{R}$ is the activation function, which is applied entry-wise. We can now write the problem as a the finite-sum:

$$\min_\theta \left\{ h(\Theta) = f(\Phi(\Theta)) = \frac{1}{m} \|V\phi(WX) - Y\|^2 = \frac{1}{m} \sum_{j=1}^m \|V\phi(WX_j) - Y_j\|^2 \right\} \tag{14}$$

We will make an assumption on the *Hermite norm* (Definition 5) of the activation function. Our assumptions on the activation function are summarized as Assumption 3 below.

**Definition 5** (Hermite norm (Olver et al., 2010))**.** *Let $\phi : \mathbb{R} \to \mathbb{R}$. The Hermite norm of $\phi$ is given by $\|\phi\|_{\mathcal{H}} = \sqrt{\sum_{i=0}^\infty c_i^2}$ where $c_i$ denotes the $i$-th Hermite coefficients of $\phi$ given by:*

$$c_i = \langle \phi, q_i \rangle_{\mathcal{H}} = \frac{1}{\sqrt{2\pi}} \int \phi(x) q_i(x) \exp\left(-\frac{x^2}{2}\right) \mathrm{d}x$$

*and $q_i : \mathbb{R} \to \mathbb{R}$ is the $i$-th Hermite polynomial (probabilist's convention) for $i \geq 0$.*

**Assumption 3.** *$\phi$ is twice-differentiable, $\phi(0) = 0$, $\sup_a |\dot\phi(a)| = \dot\phi_{\max} < \infty$, $\sup_a |\ddot\phi(a)| = \ddot\phi_{\max} < \infty$, and $\|\phi\|_{\mathcal{H}} < \infty$.*

The popular ReLU does not satisfy the twice-differentiability assumption. However, smooth approximations of ReLU such as the Gaussian error Linear Units (GeLU) and softplus (Hendrycks & Gimpel, 2020; Nguyen & Mondelli, 2020) have been shown to outperform ReLU in several settings and are commonly used in practice (Clevert et al., 2016; Gulrajani et al., 2017; Kumar et al., 2017; Kim et al., 2018; Xu et al., 2020). In addition, smoothened functions by a Gaussian kernel uniformly approximate the ReLU function (Nguyen & Mondelli, 2020).

**Assumption 4.** *For all $j \in [m]$, $\|X_j\| \leq 1$. $\|Y\| \leq 1$.*

**Remark 4.** *The assumption on the data is mild and common in the overparameterization literature (Li & Liang, 2018; Ji & Telgarsky, 2020). It can be enforced by data normalization.*

**Initialization.** The initial iterate of SGD will be chosen in the following way:

$$W^0 \sim \mathcal{N}\left(0, \frac{1}{d_0}\right), \quad V^0 \sim \mathcal{N}\left(0, \frac{1}{d_1}\right). \qquad \Theta^0 := (W^0, V^0) \tag{15}$$

**Remark 5.** *The initialization in Eq. (15) matches popular initialization schemes such as LeCun (LeCun et al., 2012) and He (He et al., 2015) initializations.*

We now proceed to estimate with high probability the value of $h(\Theta^0)$, near-isometry constants $(\mu_\Phi, \nu_\Phi)$ of $\nabla^*\Phi(\Theta^0)$ and smoothness parameter $\beta_\Phi$, which are required in Theorem 1.

**Lemma 2** (Estimation of $h(\Theta^0), \mu_\Phi, \nu_\Phi, \beta_\Phi$ (Anonymous)). *Let Assumptions 3 and 4 hold, and suppose that $\Theta^0$ follows the initialization distribution in Eq. (15). Let $t$ be a positive integer such that $m \simeq d_0^t$ and $X^{*t} \in \mathbb{R}^{d_0^t \times m}$ be the matrix whose $a$-th column defined as $\mathrm{vec}(x_a \otimes \cdots \otimes x_a) \in \mathbb{R}^{d_0^t}$.*

*For some constants $\delta_1, \delta_2, \delta_3, k_1,$ and $k_2$ independent of $d_0, d_1$ and $m$, with probability at least $1 - \tilde{\psi}$ it holds that:*

$$h(\Theta^0) \leq \frac{\delta_3^2 k_1^2 k_2^2 \sigma_{\max}^2(X)}{m}$$

$$\nu_\Phi = \max\left\{ |c_0|\sqrt{(1+\delta_2)d_1 m}, \ \omega_1\sqrt{(1+\delta_2)(c_1^2 + c_\infty^2)d_1}\sigma_{\max}(X) \right\} \tag{16}$$

$$\mu_\Phi = \sqrt{(1-\delta_1)\frac{c_t^2}{t!}d_1}\sigma_{\min}(X^{*t})$$

*The precise expression for $\tilde{\psi}$ is provided in Appendix E, $c_i$ is the $i$-th Hermite coefficients of $\phi$ (Definition 5) and $c_\infty^2 = \sum_{i=2}^\infty c_i^2/i!$.*

*Moreover, the map $\Phi$ restricted to the set $\{(V, W) : \sigma_{\max}(V) \leq \chi_{\max}\}$ is smooth with constant*

$$\beta_\Phi = \sqrt{2}\sigma_{\max}(X)\left(\dot{\phi}_{\max} + \ddot{\phi}_{\max}\chi_{\max}\right). \tag{17}$$

Although the mapping $\Phi$ is not globally smooth, Lemma 2 shows that it is smooth in a region where the largest singular value of $V$ remains bounded. In the following lemma, we show that we can indeed bound the smoothness constant of $\Phi$ restricted to a neighbourhood of $V^0$ as required in Theorem 1.

**Lemma 3.** *Let Assumption 3 hold. Let $V^0, W^0$ be arbitrary matrices and $\mu_\Phi$ be as in (16). Let*

$$\beta_\Phi := \sqrt{2}\sigma_{\max}(X)(\dot{\phi}_{\max} + \sigma_{\max}(V^0)) + \frac{\ddot{\phi}_{\max}\mu_\Phi}{2\dot{\phi}_{\max}}, \qquad \rho_\Phi := \frac{\mu_\Phi}{2\beta_\Phi} \tag{18}$$

*The function $\Phi$ is $\beta_\Phi$-smooth over the set:*

$$B_{\rho_\Phi}(V^0, W^0) := \left\{ (V, W) : \sqrt{\|V - V^0\|^2 + \|W - W^0\|^2} \leq \rho_\Phi \right\} \tag{19}$$

*Proof.* Let

$$\chi_{\max} := \sigma_{\max}(V^0) + \frac{\mu_\Phi}{2\sqrt{2}\sigma_{\max}(X)\dot{\phi}_{\max}}, \qquad \tilde{B}_{\chi_{\max}} := \{(V, W) : \sigma_{\max}(V) \leq \chi_{\max}\} \tag{20}$$

Lemma 2 then implies that $\Phi$ restricted to $\tilde{B}_{\chi_{\max}}$ is $\beta_\Phi$-smooth, following Eq. (17). With this choice of $\chi_{\max}$ we show that $B_{\rho_\Phi}(V^0, W^0) \subseteq \tilde{B}_{\chi_{\max}}$, which implies the result. Note that $\beta_\Phi \geq \sqrt{2}\sigma_{\max}(X)\dot{\phi}_{\max}$, hence

$$\chi_{\max} \geq \sigma_{\max}(V^0) + \frac{\mu_\Phi}{2\beta_\Phi} = \sigma_{\max}(V^0) + \rho_\Phi.$$

Suppose that $(V, W) \in B_{\rho_\Phi}(V^0, W^0)$. By Eq. (19) this implies $\|V - V^0\| \leq \rho_\Phi$. Then,

$$\sigma_{\max}(V) \leq \sigma_{\max}(V - V^0) + \sigma_{\max}(V^0)$$
$$\leq \|V - V^0\| + \sigma_{\max}(V^0)$$
$$\leq \rho_\Phi + \sigma_{\max}(V^0)$$
$$\leq \chi_{\max}.$$

$\square$

In our case, as $f$ is the quadratic loss, the growth condition (Definition 3) is satisfied with $\eta_f = \frac{m}{b}$. This is precisely the quantity that will reveal the impact of the minibatch size on the global convergence of SGD. Moreover, the quadratic loss satisfies the PL condition (Definition 2) and is smooth. All things considered, Assumptions 1 and 2 hold for our shallow neural network training setting, ensuring that Theorem 1 is valid.

We are now ready to integrate Lemma 2 and Lemma 3 together with the convergence guarantees in Theorem 1 to arrive at the sufficient degree of overparameterization required for the convergence of SGD. The following theorem finally concludes that for the shallow neural network described in Section 3, for a sufficient degree of overparameterization SGD converges to a global minimum with high probability.

**Theorem 2** (Shallow network with SGD). *Suppose that Assumptions 3 and 4 hold, and that that $(W^0, V^0)$ is randomly initialized as in (15). Suppose that the hidden layer width $d_1$ satisfies*

$$d_1 = \tilde{\Omega}\left(\xi(\mathcal{C}_\delta, t, \phi, \{c_i\}_{i \geq 0}, \zeta) \frac{\sigma_{\max}^2(X)m}{\sigma_{\min}^3(X^{*t})\sqrt{b}}\right) \tag{21}$$

*where $\mathcal{C}_\delta$ is a set of constants, $\xi$ is a term independent of $d_0, m$. The SGD iterates converge to a global minimum exponentially fast with probability at least $1 - \psi(\phi, \xi, d_0, d_1, d_2, X, \zeta)$. See Appendix C for the exact expressions of $\xi$ and $\psi$ and the proof.*

Finally, we provide an order analysis to understand how the sufficient overall overparameterization degree directly depends on the minibatch size. Intuitively, the sufficient overparameterization degree improves (is lower) as the minibatch size increases.

### 3.1 IMPACT OF THE MINIBATCH SIZE ON THE OVERPARAMETERIZATION DEGREE

For $t = 1$, the analysis requires $m \simeq d_0$, which is not a common setting in practice. For $t \geq 2$, we suppose that $m \simeq d_0^t$, which is the case in practice. We estimate that $\sigma_{\max}(X) \simeq \sqrt{m/d_0}$ and $\sigma_{\min}(X^{*t}) \simeq \sqrt{m/d_0^t} \simeq 1$, along the lines of (Oymak & Soltanolkotabi, 2020, Section 2.1). Substituting $\sigma_{\max}(X)$ and $\sigma_{\min}(X^{*t})$ into (21), we have

$$d_1 \gtrsim \frac{m^2}{\sqrt{b}d_0}. \tag{22}$$

Therefore, the overall overparameterization degree becomes $d_0 d_1 \simeq \tilde{\Omega}(m^2/\sqrt{b})$, which is sufficient for SGD to find a global minimum at a linear rate except with an arbitrary small probability. This fact will let us understand more clearly the effect minibatch size on the overparameterization degree.

If $b = \tilde{\Omega}(m)$, similar to gradient descent, a subquadratic scaling on the network width, $d_0 d_1 \simeq \tilde{\Omega}(m^{\frac{3}{2}})$, is sufficient. In that case, an optimal linear scaling $d_1 \simeq \tilde{O}(m)$ is sufficient when the number of input features is sufficiently large $d_0 \simeq \tilde{\Omega}(\sqrt{m})$.

On the other hand, when the batch size is small $b = \tilde{O}(1)$, we recover the standard quadratic scaling on the network width. Our analysis provides an interpolation between $d_0 d_1 \simeq \tilde{\Omega}(m^{\frac{3}{2}})$ and $d_0 d_1 \simeq \tilde{\Omega}(m^2)$ depending on $b$. As long as the batch size $b \simeq \tilde{\Omega}(m^a)$ for some $a > 0$, we achieve a subquadratic scaling.

## 4 CONCLUSIONS AND FUTURE WORK

In this work, we prove linear convergence of stochastic gradient descent for training over-parameterized two-layer neural networks with smooth activation functions, using classical initialization strategies, and where both layers are trained simultaneously. We provide a lower bound on the required over-parameterization degree for our result to hold, depending on the batch size $b$ used to compute the stochastic gradients. More precisely, we show that using a number of parameters $d_0 d_1 = \Omega(m^2/\sqrt{b})$ is sufficient to obtain linear convergence with high probability, providing sub-quadratic over-parameterization degree as long as the batch size increases with the number of data points.

In future work, we would like to relax the smoothness condition on the activation function, in order to encapsulate non-smooth activation functions such as ReLU. In addition, we would like to improve the high probability bound by analyzing more deeply the concentration properties of the random variable $\ell(I)$, characterizing the length of the trajectory spent in a neighborhood of the initialization. Finally, an important step would be to analyze the generalization properties of SGD through the lens of the proposed approach, in particular by analyzing in which case it leads to lazy training.

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

## A  PROOF OF PROPOSITION 1

According to the definition of $I$, $\mathbf{w}^i \in B$ for $i = 0, \cdots, I-1$. We first find an upper bound on the expected length of the trajectory in terms of the norm of gradients of $f$. With a proper learning rate, we find an upper bound on the norm of the gradient in terms of the expected decent of function in two consecutive iterates. We also ensure that the learning rate is sufficiently small such that $\|\mathbf{w}^I - \mathbf{w}^{I-1}\|$ is bounded. Finally, under proper initialization, the final bound is established.

The "length" of the trajectory traced by the SGD iterates $\{\mathbf{w}^i\}_{i=0}^I$ is

$$
\begin{aligned}
\ell(I) &:= \sum_{i=0}^{I-1} \|\mathbf{w}^{i+1} - \mathbf{w}^i\| \\
&= \lambda \sum_{i=0}^{I-1} \|\frac{1}{b} \sum_{j \in \mathcal{I}^i} \nabla h_j(\mathbf{w}^i)\| \\
&\lesssim \lambda \nu_\Phi \sum_{i=0}^{I-1} \|\frac{1}{b} \sum_{j \in \mathcal{I}^i} \nabla f_j(\mathbf{z}^i)\|
\end{aligned}
\tag{A.1}
$$

where $\mathbf{z}^i = \Phi(\mathbf{w}^i)$ for all $i$.

The expected length of this segment of the trajectory is therefore bounded as

$$
\begin{aligned}
\mathbb{E}[\ell(I)] &\lesssim \lambda \nu_\Phi \mathbb{E}\left[ \sum_{i=0}^{I-1} \|\frac{1}{b} \sum_{j \in \mathcal{I}^i} \nabla f_j(\mathbf{z}^i)\| \right] \\
&\leq \lambda \sqrt{\eta_f} \nu_\Phi \mathbb{E}\left[ \sum_{i=0}^{I-1} \|\nabla f(\mathbf{z}^i)\| \right]
\end{aligned}
\tag{A.2}
$$

where the last inequality holds thanks to the growth condition and Jensen's inequality.

We now develop a "descent inequality' and establish a lower bound on $f(\mathbf{z}^i) - \mathbb{E}[f(\mathbf{z}^{i+1})|\mathbf{w}^i]$. First, consider two consecutive SGD iterates $i, i+1$ such that $\mathbf{w}^i, \mathbf{w}^{i+1} \in B$. Then, we have

$$
\begin{aligned}
f(\mathbf{z}^i) - f(\mathbf{z}^{i+1}) &\geq \langle \mathbf{z}^i - \mathbf{z}^{i+1}, \nabla f(\mathbf{z}^i) \rangle - \frac{\beta_f}{2} \|\mathbf{z}^{i+1} - \mathbf{z}^i\|^2 \\
&= \langle \Phi(\mathbf{w}^i) - \Phi(\mathbf{w}^{i+1}), \nabla f(\mathbf{z}^i) \rangle - \frac{\beta_f}{2} \|\Phi(\mathbf{w}^{i+1}) - \Phi(\mathbf{w}^i)\|^2 \\
&= \langle \nabla \Phi(\mathbf{w}^i)\left\{ \mathbf{w}^i - \mathbf{w}^{i+1} \right\}, \nabla f(\mathbf{z}^i) \rangle - \frac{\beta_f}{2} \|\Phi(\mathbf{w}^{i+1}) - \Phi(\mathbf{w}^i)\|^2 \\
&\quad - \langle \Phi(\mathbf{w}^{i+1}) - \Phi(\mathbf{w}^i) - \nabla \Phi(\mathbf{w}^i)\left\{ \mathbf{w}^{i+1} - \mathbf{w}^i \right\}, \nabla f(\mathbf{z}^i) \rangle \\
&\geq \langle \nabla \Phi(\mathbf{w}^i)\left\{ \mathbf{w}^i - \mathbf{w}^{i+1} \right\}, \nabla f(\mathbf{z}^i) \rangle - \frac{\beta_f}{2} \|\Phi(\mathbf{w}^{i+1}) - \Phi(\mathbf{w}^i)\|^2
\end{aligned}
$$

$$- \frac{\beta_\Phi}{2} \|\mathbf{w}^{i+1} - \mathbf{w}^i\|^2 \|\nabla f(\mathbf{z}^i)\|$$

$$\geq \langle \nabla\Phi(\mathbf{w}^i)\left\{\mathbf{w}^i - \mathbf{w}^{i+1}\right\}, \nabla f(\mathbf{z}^i)\rangle - \frac{1}{2}\|\mathbf{w}^{i+1} - \mathbf{w}^i\|^2 \left(\beta_\Phi\|\nabla f(\mathbf{z}^i)\| + \frac{9\beta_f \nu_\Phi^2}{4}\right)$$

$$= \lambda\langle \nabla\Phi(\mathbf{w}^i)\{\frac{1}{b}\sum_{j\in\mathcal{I}^i}\nabla h_j(\mathbf{w}^i)\}, \nabla f(\mathbf{z}^i)\rangle - \frac{\lambda^2}{2}\|\frac{1}{b}\sum_{j\in\mathcal{I}^i}\nabla h_j(\mathbf{w}^i)\|^2 \left(\beta_\Phi\|\nabla f(\mathbf{z}^i)\| + \frac{9\beta_f\nu_\Phi^2}{4}\right)$$

$$\text{(A.3)}$$

Hence, taking expectation over the selected batch $n^i$ at iteration $i$, we find that

$$f(\mathbf{z}^i) - \mathbb{E}[f(\mathbf{z}^{i+1})|\mathbf{w}^i]$$

$$\geq \mathbb{E}\left[\left(\lambda\langle\nabla\Phi(\mathbf{w}^i)\{\frac{1}{b}\sum_{j\in\mathcal{I}^i}\nabla h_j(\mathbf{w}^i)\}, \nabla f(\mathbf{z}^i)\rangle - \frac{\lambda^2}{2}\|\frac{1}{b}\sum_{j\in\mathcal{I}^i}\nabla h_j(\mathbf{w}^i)\|\left(\beta_\Phi\|\nabla f(\mathbf{z}^i)\| + \frac{9\beta_f\nu_\Phi^2}{4}\right)\right)|\mathbf{w}^i\right]$$

$$= \left(\lambda\langle\nabla\Phi(\mathbf{w}^i)\{\nabla h(\mathbf{w}^i)\}, \nabla f(\mathbf{z}^i)\rangle - \frac{\lambda^2}{2}\mathbb{E}\left[\|\frac{1}{b}\sum_{j\in\mathcal{I}^i}\nabla h_j(\mathbf{w}^i)\|^2|\mathbf{w}^i\right]\left(\beta_\Phi\|\nabla f(\mathbf{z}^i)\| + \frac{9\beta_f\nu_\Phi^2}{4}\right)\right)$$

$$= \left(\lambda\|\nabla h(\mathbf{w}^i)\|^2 - \frac{\lambda^2}{2}\mathbb{E}\left[\|\frac{1}{b}\sum_{j\in\mathcal{I}^i}\nabla h_j(\mathbf{w}^i)\|^2|\mathbf{w}^i\right]\left(\beta_\Phi\|\nabla f(\mathbf{z}^i)\| + \frac{9\beta_f\nu_\Phi^2}{4}\right)\right)$$

$$\geq \left(\lambda\|\nabla h(\mathbf{w}^i)\|^2 - \frac{\lambda^2\eta_f\nu_\Phi^2}{2}\|\nabla f(\mathbf{z}^i)\|^2\left(\beta_\Phi\|\nabla f(\mathbf{z}^i)\| + \frac{9\beta_f\nu_\Phi^2}{4}\right)\right)$$

$$= \lambda\mu_\Phi^2\|\nabla f(\mathbf{z}^i)\|^2\left(1 - \frac{\lambda\eta_f\nu_\Phi^2\beta_\Phi\|\nabla f(\mathbf{z}^i)\|}{2\mu_\Phi^2} - \frac{9\eta_f\beta_f\nu_\Phi^4}{8\mu_\Phi^2}\right) \quad \text{(chain rule and Lemma 1)}$$

$$\gtrsim \lambda\mu_\Phi^2\|\nabla f(\mathbf{z}^i)\|^2$$

where the final inequality holds if the learning rate is small enough such that

$$\lambda \lesssim \frac{\mu_\Phi^2}{\eta_f\beta_\Phi\nu_\Phi^2\max_i\|\nabla f(\mathbf{z}^i)\| + \eta_f\beta_f\nu_\Phi^4}. \tag{A.4}$$

We also note that:

$$\sqrt{f(\mathbf{z}^i)} - \sqrt{\mathbb{E}[f(\mathbf{z}^{i+1})|\mathbf{w}^i]} = \frac{(f(\mathbf{z}^i) - \mathbb{E}[f(\mathbf{z}^{i+1})|\mathbf{w}^i])}{\sqrt{f(\mathbf{z}^i)} + \sqrt{\mathbb{E}[f(\mathbf{z}^{i+1})|\mathbf{w}^i]}}$$

$$\gtrsim \frac{\lambda\mu_\Phi^2\|\nabla f(\mathbf{z}^i)\|^2}{\sqrt{f(\mathbf{z}^i)} + \sqrt{\mathbb{E}[f(\mathbf{z}^{i+1})|\mathbf{w}^i]}}$$

$$\geq \frac{\lambda\mu_\Phi^2\|\nabla f(\mathbf{z}^i)\|^2}{2\sqrt{f(\mathbf{z}^i)}} \tag{A.5}$$

$$\geq \frac{\lambda\sqrt{\alpha_f}\mu_\Phi^2\|\nabla f(\mathbf{z}^i)\|^2}{\sqrt{2}\|\nabla f(\mathbf{z}^i)\|}$$

$$= \frac{\lambda\sqrt{\alpha_f}\mu_\Phi^2}{\sqrt{2}}\|\nabla f(\mathbf{z}^i)\|.$$

Substituting the above back into (A.2), and using the fact that $\mathbf{w}^i \in B \; \forall i = 0, \dots, I-1$, an upper bound on $\mathbb{E}[\ell(I)]$ is given by:

$$\mathbb{E}[\ell(I)] \leq \lambda\sqrt{\eta_f}\nu_\Phi\mathbb{E}\left[\sum_{i=0}^{I-1}\|\nabla f(\mathbf{z}^i)\|\right]$$

$$\lesssim \frac{\sqrt{\eta_f}\nu_\Phi}{\sqrt{\alpha_f}\mu_\Phi^2}\mathbb{E}\left[\sum_{i=0}^{I-2}\left(\sqrt{f(\mathbf{z}^i)} - \sqrt{\mathbb{E}[f(\mathbf{z}^{i+1})|\mathbf{w}^i]}\right)\right] + \lambda\sqrt{\eta_f}\nu_\Phi\mathbb{E}\left[\|\nabla f(\mathbf{z}^{I-1})\|\right]$$

$$\leq \frac{\sqrt{\eta_f}\nu_\Phi}{\sqrt{\alpha_f}\mu_\Phi^2}\mathbb{E}\left[\sum_{i=0}^{I-1}\sqrt{f(\mathbf{z}^i)} - \sqrt{f(\mathbf{z}^{i+1})}\right] + \lambda\sqrt{\eta_f}\nu_\Phi\mathbb{E}\left[\|\nabla f(\mathbf{z}^{I-1})\|\right] \qquad \text{(Jensen's inequality)}$$

$$= \frac{\sqrt{\eta_f}\nu_\Phi}{\sqrt{\alpha_f}\mu_\Phi^2}\mathbb{E}\left[\sqrt{f(\mathbf{z}^0)} - \sqrt{f(\mathbf{z}^I)}\right] + \lambda\sqrt{\eta_f}\nu_\Phi\mathbb{E}\left[\|\nabla f(\mathbf{z}^{I-1})\|\right]$$

$$\leq \frac{\sqrt{\eta_f}\nu_\Phi\sqrt{f(\mathbf{z}^0)}}{\sqrt{\alpha_f}\mu_\Phi^2} + \lambda\sqrt{\eta_f}\nu_\Phi\mathbb{E}\left[\|\nabla f(\mathbf{z}^{I-1})\|\right]. \tag{A.6}$$

Suppose that, in addition to the previous bound on the learning rate, we have

$$\lambda \leq \frac{\mu_\Phi}{4\zeta\sqrt{\eta_f}\nu_\Phi\beta_\Phi\max_i\|\nabla f(\mathbf{z}^i)\|}, \tag{A.7}$$

and that the initialization satisfies

$$f(\mathbf{z}^0) = h(\mathbf{w}^0) \lesssim \frac{\alpha_f\mu_\Phi^6}{\zeta^2\beta_f^2\eta_f\nu_\Phi^2}. \tag{A.8}$$

for some $\zeta > 1$. Then, we obtain (12).:

$$\mathbb{E}[\ell(I)] \leq \frac{\mu_\Phi}{2\zeta\beta_\Phi} = \frac{\rho_\Phi}{\zeta}.$$

We note that the local Lipschitz constant of $f$ is given by

$$\begin{aligned}
\max_i\|\nabla f(\mathbf{z}^i)\| &\leq \|\nabla f(\mathbf{z}^0)\| + \max_i\|\nabla f(\mathbf{z}^i) - \nabla f(\mathbf{z}^0)\| \\
&\leq \|\nabla f(\mathbf{z}^0)\| + \beta_f\max_i\|\mathbf{z}^i - \mathbf{z}^0\| \\
&= \|\nabla f(\mathbf{z}^0)\| + \beta_f\max_i\|\Phi(\mathbf{w}^i) - \Phi(\mathbf{w}^0)\| \\
&= \|\nabla f(\mathbf{z}^0)\| + \frac{3\beta_f\nu_\Phi}{2}\max_i\|\mathbf{w}^i - \mathbf{w}^0\| \\
&\leq \|\nabla f(\mathbf{z}^0)\| + \frac{3\beta_f\nu_\Phi}{2}\cdot\rho_\Phi \\
&= \|\nabla f(\mathbf{z}^0)\| + \frac{3\beta_f\mu_\Phi\nu_\Phi}{4\beta_\Phi}.
\end{aligned} \tag{A.9}$$

Substituting (A.9) into (A.4) and (A.7), an upper bound on the learning rate of SGD is given by

$$\lambda \lesssim \min\left(\frac{\mu_\Phi^2}{\eta_f(\beta_\Phi\nu_\Phi^2\|\nabla f(\mathbf{z}^0)\| + \beta_f\nu_\Phi^4 + \beta_f\mu_\Phi\nu_\Phi^3)}, \frac{\mu_\Phi}{\zeta\sqrt{\eta_f}\nu_\Phi(\beta_\Phi\|\nabla f(\mathbf{z}^0)\| + \beta_f\nu_\Phi\mu_\Phi)}\right).$$

This completes the proof of Proposition 1.

## B    PROOF THEOREM 1

Let $\zeta > 1$. Using Proposition 1, under proper initialization and choice of learning rate, the expected length of SGD trajectory is bounded above by:

$$\mathbb{E}[\ell(I)] \leq \frac{\mu_\Phi}{2\zeta\beta_\Phi} = \frac{\rho_\Phi}{\zeta}.$$

Thus, using Markov inequality on $\ell(I)$, we have

$$\Pr\{\ell(I) \geq \rho_\Phi\} \leq \frac{1}{\zeta}.$$

Therefore, with probability greater than $1 - \frac{1}{\zeta}$, we have $\ell(I) < \rho_\Phi$, meaning that the SGD trajectory never leaves the ball $B$, *i.e.,* $I = \infty$. Moreover, conditioned on this event, $\ell(I)$ corresponds to the length of the full trajectory. Since we have a bound on $\mathbb{E}[\ell(I)]$, it implies that the trajectory length is finite, i.e., that the SGD iterates converge.

Finally, we show the linear convergence to the optimal limit point:

$$
\begin{aligned}
\mathbb{E}[h(\mathbf{w}^{i+1})|\mathbf{w}^i] &= \mathbb{E}[h(\mathbf{w}^{i+1})|\mathbf{w}^i] - h(\mathbf{w}^i) + h(\mathbf{w}^i) \\
&= \mathbb{E}[f(\mathbf{z}^{i+1})|\mathbf{w}^i] - f(\mathbf{z}^i) + h(\mathbf{w}^i) \\
&\leq -C\lambda\mu_\Phi^2\|\nabla f(\mathbf{z}^i)\|^2 + h(\mathbf{w}^i) \\
&\leq -C\lambda\alpha_f\mu_\Phi^2 f(\mathbf{z}^i) + h(\mathbf{w}^i) \\
&= (1 - C\lambda\alpha_f\mu_\Phi^2)h(\mathbf{w}^i),
\end{aligned}
\tag{A.10}
$$

where $C$ denotes constants and $\mathbf{z}^i = \Phi(\mathbf{w}^i)$ for all $i$. Taking expectation of both sides of (A.10) w.r.t. $n^0, \cdots, n^{i-1}$ completes the proof of Theorem 1.

## C  PROOF OF THEOREM 2

For the case of quadratic loss functions, we have $\alpha_f = \beta_f = \frac{2}{m}$. In the following, we first find the growth condition parameter:

Recall $f_j(Z) = \|Z_j - Y_j\|^2$. Although $\nabla f_j(Z)$ is a matrix is a matrix with the same size as $Z$, i.e., $\nabla f(Z) \in \mathbb{R}^{d_2 \times m}$, $f_j$ only depends on the $j^{th}$ column of $Z$, and hence only the $j^{th}$ column of $\nabla f_j(Z)$ will be non zero. More precisely, we have

$$
(\nabla f_j(Z))_{\cdot j} = \frac{2}{b}(Z_j - Y_j).
$$

and $(\nabla f_j(Z))_{\cdot k} = 0$ for $k \neq j$, where $(\nabla f_j(Z))_{\cdot k}$ denotes the $k^{th}$ column of $\nabla f_j(Z)$. We thus have that

$$
\begin{aligned}
\mathbb{E}_{\mathcal{I},|\mathcal{I}|=b}\left[\|\frac{1}{b}\sum_{j\in\mathcal{I}}\nabla f_j(Z)\|^2\right] &= \frac{4}{b}\mathbb{E}_{\mathcal{I},|\mathcal{I}|=b}\left[\frac{1}{b}\sum_{j\in\mathcal{I}}\|Z_j - Y_j\|^2\right] \\
&= \frac{4}{b}\mathbb{E}[f_j(Z)] \\
&= \frac{4}{b}f(Z).
\end{aligned}
$$

Similarly, we have $\|\nabla f(Z)\|^2 = \frac{4}{m}f(Z)$. Hence, the quadratic loss satisfies growth condition with $\eta_f = \frac{m}{b}$.

Suppose there exists some constant $t \geq 1$ such that $m \simeq d_0^t$.

Using Lemma 2 and Lemma 3, we can estimate the parameters $\nu_\Phi, \mu_\Phi, \beta_\Phi$ and establish an upper bound on $h(\Theta^0)$:

$$
\begin{aligned}
h(\Theta^0) &\leq \frac{1}{m}\delta_3^2 k_1^2 k_2^2 \sigma_{\max}^2(X) \\
\nu_\Phi &= \max\left\{|c_0|\sqrt{(1+\delta_2)d_1 m}, \ \omega_1\sqrt{(1+\delta_2)(c_1^2 + c_\infty^2)d_1}\sigma_{\max}(X)\right\} \\
\mu_\Phi &= \sqrt{(1-\delta_1)\frac{c_t^2}{t!}d_1\sigma_{\min}(X^{*t})} \\
\beta_\Phi &= \sqrt{2}\sigma_{\max}(X)(\dot{\phi}_{\max} + \sigma_{\max}(V^0)) + \frac{\ddot{\phi}_{\max}\mu_\Phi}{2\dot{\phi}_{\max}}
\end{aligned}
\tag{A.11}
$$

where $\delta_1, \delta_2, \ \delta_3, k_1,$ and $k_2$ are all constants and independent of $d_0, \ d_1,$ and $m$; the term $t$ is a constant such that $m \simeq d_0^t$, and $X^{*t} \in \mathbb{R}^{d_0^t \times m}$ is derived from Khatri-Rao product with its $a$-th column defined as $\text{vec}(x_a \otimes \cdots \otimes x_a) \in \mathbb{R}^{d_0^t}$, with probability at least $1 - \tilde{\psi}$ where $\tilde{\psi} =$

$$d_1^{-C\delta_4 d_0} - d_1^{-C\delta_4 d_2} - e^{-\left(\frac{\delta_1 \sigma_{\min}(\mathbb{E}[M^0])}{4\dot{\phi}_{\max}^2 \sigma_{\max}^2(X)\delta_4\sqrt{d_0 \log d_1}}\right)^2} - e^{-\left(\frac{\delta_2 \sigma_{\max}(\mathbb{E}[M^0])}{4\dot{\phi}_{\max}^2 \sigma_{\max}^2(X)\delta_4\sqrt{d_0 \log d_1}}\right)^2} - e^{-Cd_1} - e^{-C\delta_3^2},$$

$\delta_4 = \max(k_1, k_2)$, $M_0 = \phi(X^\top W^{0\top})\phi(W^0 X)$, $C$ is a constant, and

$$\omega_1^{2t} d_1 \frac{c_t^2}{t!}\sigma_{\min}^2(X^{*t}) \lesssim \sigma_{\min}(\mathbb{E}[M^0]) \leq \sigma_{\max}(\mathbb{E}[M^0]) \lesssim d_1 \left(mc_0^2 + \omega_1^2(c_1^2 + c_\infty^2)\sigma_{\max}^2(X)\right).$$

We denote $\mathcal{C}_\delta = \{\delta_1, \delta_2, \delta_3, \delta_4\}$.

We highlight the main condition for the linear convergence of SGD in (6):

$$h(\Theta^0) = O\left(\frac{\alpha_f \mu_\Phi^6}{\zeta \beta_\Phi^2 \eta_f \nu_\Phi^2}\right).$$

The convergence happens with probability at least $1 - \frac{1}{\zeta}$.

Suppose $|c_0|$ is sufficiently large such that $|c_0|\sqrt{(1+\delta_2)d_1 m}$ becomes the dominating term in $\nu_\Phi$.[1]

We note that the order of $\sigma_{\max}(X)$ and $\sigma_{\min}(X^{*t})$ play significant roles for the overparameterization order analysis. For $t = 1$, it requires $m \simeq d_0$, which is not a common setting in practice. In the following, we focus on $t \geq 2$.

Substituting the parameters obtained in (A.11) into (6), a sufficient condition to satisfy (6) is given by

$$d_1 \gtrsim \sqrt{\xi(\mathcal{C}_\delta, t, \phi, \{c_i\}_{i \geq 0}, \zeta)} \cdot \frac{m\sigma_{\max}^2(X)}{\sqrt{b}\sigma_{\min}^3(X^{*t})}.$$

where

$$\xi(\mathcal{C}_\delta, t, \phi, \{c_i\}_{i \geq 0}, \zeta) = \sqrt{\frac{\zeta\delta_3^2 c_0^2(1+\delta_2)\omega_1^{4t+4}\delta_4^4(\dot{\phi}_{\max} + \sigma_{\max}(V^0))^2 t!^3}{(1-\delta_1)^3 c_t^6}}.$$

Since $\xi(\mathcal{C}_\delta, t, \phi, \{c_i\}_{i \geq 0}, \zeta)$ is constant w.r.t. $d_0$, $d_1$, and $m$. The sufficient condition can be expressed as:

$$d_1 = \tilde{\Omega}\left(\frac{m\sigma_{\max}^2(X)}{\sqrt{b}\sigma_{\min}^3(X^{*t})}\right). \tag{A.12}$$

Finally, along the lines of (Oymak & Soltanolkotabi, 2020)[Section 2.1], we have $\sigma_{\max}(X) \simeq \sqrt{\frac{m}{d_0}}$ and $\sigma_{\min}(X^{*t}) \simeq \sqrt{\frac{m}{d_0^t}} \simeq 1$.

Replacing $\sigma_{\max}(X)$ and $\sigma_{\min}(X^{*t})$ into (A.12), the overall overparameterization degree becomes $d_0 d_1 \simeq \tilde{\Omega}(m^2/\sqrt{b})$. Finally using the union bound on the events corresponding to random initialization and random SGD iterates, the linear convergence is established with probability at least $1 - \psi$ where

$$\psi = d_1^{-C\delta_4 d_0} + d_1^{-C\delta_4 d_2} + e^{-Cd_1} + e^{-C\delta_3^2} + \frac{1}{\zeta}$$
$$+ e^{-\left(\frac{\delta_1 \sigma_{\min}(\mathbb{E}[M^0])}{4\dot{\phi}_{\max}^2 \sigma_{\max}^2(X)\delta_4\sqrt{d_0 \log d_1}}\right)^2} + e^{-\left(\frac{\delta_2 \sigma_{\max}(\mathbb{E}[M^0])}{4\dot{\phi}_{\max}^2 \sigma_{\max}^2(X)\delta_4\sqrt{d_0 \log d_1}}\right)^2}. \tag{A.13}$$

## D  PROOF OF LEMMA 1

The content of this proof is originally due to (Anonymous) and provided here for the completeness of our paper.

Intuitively, if $\nabla\Phi^*(\mathbf{w}^0)$ is a $(\mu_\Phi, \nu_\Phi)$-near-isometry, then one would expect $\nabla\Phi^*$ to remain near-isometry for all nearby points. Formally, let $A, B \in R^{m \times n}$ and let singular values of a matrix are ordered such that $\sigma_i(A) \geq \sigma_j(A)$ and $\sigma_i(B) \geq \sigma_j(B)$ for $1 \leq i \leq j \leq \min\{m, n\}$. Using Weyl's inequality and for $i + j - 1 \leq \min\{m, n\}$, we have:

---

[1]The same scaling holds if the other term dominates.

$$\sigma_{i+j-1}(A+B) \leq \sigma_i(A) + \sigma_j(B). \tag{A.14}$$

More formally, suppose that $\mathbf{w} \in \mathbb{R}^d$ satisfies

$$\|\mathbf{w} - \mathbf{w}^0\| \leq \frac{\mu_\Phi}{2\beta_\Phi} = \rho_\Phi. \tag{A.15}$$

If $\nabla\Phi^*(\mathbf{w}^0)$ is $(\mu_\Phi, \nu_\Phi)$-isometry in the sense of Definition 4, then applying Weyl's inequality (A.14) along with using smoothness and (A.15), we have

$$\begin{aligned}
\sigma_{\min}(\nabla\Phi^*(\mathbf{w})) &\geq \sigma_{\min}(\nabla\Phi^*(\mathbf{w}^0)) - \sigma_{\max}(\nabla\Phi^*(\mathbf{w}) - \nabla\Phi^*(\mathbf{w}^0)) \\
&\geq \mu_\Phi - \beta_\Phi\|\mathbf{w} - \mathbf{w}^0\| \\
&\geq \frac{\mu_\Phi}{2}.
\end{aligned}$$

Using a similar argument, we establish an upper bound $\sigma_{\max}(\nabla\Phi^*(\mathbf{w}))$:

$$\sigma_{\max}(\nabla\Phi^*(\mathbf{w})) \leq \sigma_{\max}(\nabla\Phi^*(\mathbf{w}^0)) + \sigma_{\max}(\nabla\Phi^*(\mathbf{w}) - \nabla\Phi^*(\mathbf{w}^0)) \leq \nu_\Phi + \frac{\mu_\Phi}{2} \leq \frac{3\nu_\Phi}{2}.$$

# E   PROOF OF LEMMA 2

**Lemma A.4** (Estimation of $h(\Theta^0), \mu_\Phi, \nu_\Phi, \beta_\Phi$ (Anonymous)). *Suppose that the shallow neural network satisfies Assumption 3. Then we have*

$$\begin{aligned}
h(\Theta^0) &\leq \frac{1}{m}\delta_3^2 k_1^2 k_2^2 \sigma_{\max}^2(X) \\
\nu_\Phi &= \max\left\{|c_0|\sqrt{(1+\delta_2)d_1 m},\ \omega_1\sqrt{(1+\delta_2)(c_1^2 + c_\infty^2)d_1}\sigma_{\max}(X)\right\} \\
\mu_\Phi &= \sqrt{(1-\delta_1)\frac{c_t^2}{t!}d_1\sigma_{\min}(X^{*t})} \\
\beta_\Phi &= \sqrt{2}\sigma_{\max}(X)\left(\dot{\phi}_{\max} + \ddot{\phi}_{\max}\chi_{\max}\right).
\end{aligned} \tag{A.16}$$

*where $\delta_1, \delta_2, \delta_3, k_1,$ and $k_2$ are all constants and independent of $d_0$, $d_1$, and $m$; $\beta_\Phi$ is the smoothness constant of the map $\Phi$ restricted to the set $\{(V, W) : \sigma_{\max}(V) \leq \chi_{\max}\}$; the term $t$ is a constant such that $m \simeq d_0^t$, and $X^{*t} \in \mathbb{R}^{d_0^t \times m}$ is derived from Khatri-Rao product with its $a$-th column defined as $\mathrm{vec}(x_a \otimes \cdots \otimes x_a) \in \mathbb{R}^{d_0^t}$, with probability at least $1 - \tilde{\psi}$ where $\tilde{\psi} = d_1^{-C\delta_4 d_0} - d_1^{-C\delta_4 d_2} - e^{-\left(\frac{\delta_1\sigma_{\min}(\mathbb{E}[M^0])}{4\dot{\phi}_{\max}^2\sigma_{\max}^2(X)\delta_4\sqrt{d_0\log d_1}}\right)^2} - e^{-\left(\frac{\delta_2\sigma_{\max}(\mathbb{E}[M^0])}{4\dot{\phi}_{\max}^2\sigma_{\max}^2(X)\delta_4\sqrt{d_0\log d_1}}\right)^2} - e^{-Cd_1} - e^{-C\delta_3^2}$, where $\delta_4 = \max(k_1, k_2)$, $M_0 = \phi(X^\top W^{0\top})\phi(W^0 X)$, $C$ is a constant, and*

$$\omega_1^{2t}d_1\frac{c_t^2}{t!}\sigma_{\min}^2(X^{*t}) \lesssim \sigma_{\min}(\mathbb{E}[M^0]) \leq \sigma_{\max}(\mathbb{E}[M^0]) \lesssim d_1\left(mc_0^2 + \omega_1^2(c_1^2 + c_\infty^2)\sigma_{\max}^2(X)\right),$$

*. We denote $\mathcal{C}_\delta = \{\delta_1, \delta_2, \delta_3, \delta_4\}$.*

The content of this proof is originally due to (Anonymous) and provided here for the completeness of our paper.

We first obtain the expression for adjoint operator $\nabla\Phi^*(\Theta) : \mathbb{R}^{d_2 \times m} \to \mathbb{R}^{d_1 \times d_0} \times \mathbb{R}^{d_2 \times d_1}$. Let $\Delta_W \in \mathbb{R}^{d_1 \times d_0}, \Delta_V \in \mathbb{R}^{d_2 \times d_1}$, and $\Delta \in \mathbb{R}^{d_2 \times m}$. We expand $\Phi$ as follow:

$$\begin{aligned}
\Phi(W + \Delta_W, V) &\approx \Phi(W, V) + \nabla_W\Phi(\Delta_W), \\
\Phi(W, V + \Delta_V) &\approx \Phi(W, V) + \nabla_V\Phi(\Delta_V)
\end{aligned} \tag{A.17}$$

where

$$\nabla_W\Phi(\Delta_W) = V\left(\dot{\phi}(WX) \odot \Delta_W X\right), \quad \nabla_V\Phi(\Delta_V) = \Delta_V\phi(WX),$$

$\odot$ stands for the Hadamard (entry-wise) product, and $\dot{\phi}(WX)$ is the derivative of $\phi$ calculated at each entry of the matrix $WX$. The operator $\nabla\Phi(\Theta)$ is given by $(\Delta_W, \Delta_V) \to \nabla_W\Phi(\Delta_W) + \nabla_V\Phi(\Delta_V)$.

Using the cyclic property of the $\text{trace}$ operator and $\text{trace}\left((A \odot B)C\right) = \text{trace}\left((A \odot C^\top)B^\top\right)$, we have

$$
\begin{aligned}
\langle \Delta, \nabla_W\Phi(\Delta_W)\rangle &= \left\langle \left(\dot{\phi}(WX) \odot V^\top\Delta\right)X^\top, \Delta_W \right\rangle, \\
\langle \Delta, \nabla_V\Phi(\Delta_V)\rangle &= \left\langle \Delta_V, \Delta\phi\left(X^\top W^\top\right)\right\rangle.
\end{aligned}
\tag{A.18}
$$

Substituting (A.18), the adjoint operator is given by

$$
\nabla\Phi^*(\Theta) : \Delta \to \left(\left(\dot{\phi}(WX) \odot V^\top\Delta\right)X^\top, \Delta\phi\left(X^\top W^\top\right)\right).
\tag{A.19}
$$

Suppose that there exist $\dot{\phi}_{\max}, \ddot{\phi}_{\max} < \infty$ such that

$$
\sup_a |\dot{\phi}(a)| \le \dot{\phi}_{\max}, \quad \sup_a |\ddot{\phi}(a)| \le \ddot{\phi}_{\max}.
\tag{A.20}
$$

**Lemma A.5.** *Let $A \in \mathbb{R}^{m\times n}$ and $B \in \mathbb{R}^{n\times k}$. Then, we have*

$$
\sigma_{\min}(A)\|B\| \le \|AB\| \le \sigma_{\max}(A)\|B\|.
$$

Using Lemma A.5 and triangular inequality, we note that

$$
\begin{aligned}
\|\nabla\Phi^*(\Theta, \Delta)\| &\le \left\|\left(\dot{\phi}(WX) \odot (V^\top\Delta)\right)X^\top\right\| + \|\Delta\phi(X^\top W^\top)\| \\
&\le \dot{\phi}_{\max}\sigma_{\max}(X)\sigma_{\max}(V)\|\Delta\| + \sigma_{\max}(\phi(WX))\|\Delta\|.
\end{aligned}
\tag{A.21}
$$

Similarly, we have this lower bound:

$$
\|\nabla\Phi^*(\Theta, \Delta)\| \ge \sigma_{\min}(\phi(WX))\|\Delta\|.
\tag{A.22}
$$

Substituting $\Theta_0 = (W_0, V_0)$ into (A.21) and (A.22), $\mu_\Phi$ and $\nu_\Phi$ are given by:

$$
\begin{aligned}
\sigma_{\max}(\nabla\Phi^*(\Theta_0)) &\le \dot{\phi}_{\max}\sigma_{\max}(X)\sigma_{\max}(V_0) + \sigma_{\max}(\phi(W_0X)) =: \nu_\Phi, \\
\sigma_{\min}(\nabla\Phi^*(\Theta_0)) &\ge \sigma_{\min}(\phi(W_0X)) =: \mu_\Phi.
\end{aligned}
\tag{A.23}
$$

**Lemma A.6.** *Let $A \in \mathbb{R}^{m\times n}$ and $B \in \mathbb{R}^{n\times k}$. Then, for $p, q \ge 1$, we have*

$$
\|AB\|_q \le \|A\|_p\|B\|_q.
$$

In the following, we find the smoothness parameter $\beta_\Phi$ in (1). Let $\Theta, \hat{\Theta} \in \mathbb{R}^{d_1\times d_0} \times \mathbb{R}^{d_2\times d_1}$. We note that $\|\nabla\Phi(\Theta, \Delta) - \nabla\Phi(\hat{\Theta}, \Delta)\| \le U_1 + U_2$ where

$$
\begin{aligned}
U_1 &= \|V(\dot{\phi}(W^\top X) \odot (\Delta_W^\top X)) - \hat{V}(\dot{\phi}(\hat{W}^\top X) \odot (\Delta_W^\top X))\| \\
U_2 &= \|\Delta_V\phi(W^\top X) - \Delta_V\phi(\hat{W}^\top X)\|.
\end{aligned}
\tag{A.24}
$$

Let us denote

$$
\sigma_{\max}(\hat{V}) \le \chi_{\max}.
\tag{A.25}
$$

An upper bound on $U_1$ in (A.24) is given by:

$$
\begin{aligned}
U_1 &\le \|(V - \hat{V})(\dot{\phi}(W^\top X) \odot (\Delta_W^\top X))\| + \|\hat{V}(\dot{\phi}(W^\top X) \odot (\Delta_W^\top X) - \hat{V}\dot{\phi}(\hat{W}^\top X) \odot (\Delta_W^\top X))\| \\
&\le \dot{\phi}_{\max}\sigma_{\max}(X)\|V - \hat{V}\|\|\Delta_W\| + \sigma_{\max}(X)\sigma_{\max}(\hat{V})\|\dot{\phi}(W^\top X) - \dot{\phi}(\hat{W}^\top X)\|_\infty\|\Delta_W\| \\
&\le \dot{\phi}_{\max}\sigma_{\max}(X)\|V - \hat{V}\|\|\Delta_W\| + \ddot{\phi}_{\max}\sigma_{\max}(X)\|X\|_\infty\sigma_{\max}(\hat{V})\|W - \hat{W}\|\|\Delta_W\| \\
&\le \dot{\phi}_{\max}\sigma_{\max}(X)\|V - \hat{V}\|\|\Delta_W\| + \ddot{\phi}_{\max}\chi_{\max}\sigma_{\max}(X)\|W - \hat{W}\|\|\Delta_W\|
\end{aligned}
$$

where the second last inequality is due to Lemma A.6.

An upper bound on $U_2$ in (A.24) is given by:

$$U_2 \leq \dot{\phi}_{\max}\sigma_{\max}(X)\|W - \hat{W}\|\|\Delta_V\|.$$

Substituting the upper bounds on $U_1$ and $U_2$, an upper bound on $\sigma_{\max}(\nabla\Phi(\Theta) - \nabla\Phi(\hat{\Theta}))$ is given by

$$\sigma_{\max}(\nabla\Phi(\Theta) - \nabla\Phi(\hat{\Theta})) \leq \sigma_{\max}(X)\left(\dot{\phi}_{\max} + \ddot{\phi}_{\max}\chi_{\max}\right)\|W - \hat{W}\| + \sigma_{\max}(X)\dot{\phi}_{\max}\|V - \hat{V}\|$$

$$\leq \sqrt{2}\sigma_{\max}(X)\left(\dot{\phi}_{\max} + \ddot{\phi}_{\max}\chi_{\max}\right)\|\Theta - \hat{\Theta}\|$$

where the last inequality holds since

$$\|W - \hat{W}\| + \|V - \hat{V}\| \leq \sqrt{2}\sqrt{\|W - \hat{W}\|^2 + \|V - \hat{V}\|^2}.$$

Finally, $\beta_\Phi$ in (1) is given by

$$\beta_\Phi = \sqrt{2}\sigma_{\max}(X)\left(\dot{\phi}_{\max} + \ddot{\phi}_{\max}\chi_{\max}\right). \tag{A.26}$$

### E.1 ESTIMATING $\mu_\Phi, \nu_\Phi$

We now estimate the random quantities $\mu_\Phi, \nu_\Phi$ in our neural network setting. They key quantities to estimate are $\sigma_{\min}(\phi(W^0X))$ and $\sigma_{\max}(\phi(W^0X))$. To that end, we consider Hermite decomposition of the activation function $\phi$.

We start with the basic definition of Hermite polynomial and its properties. Let $i \geq 0$ and let $q_i : \mathbb{R} \to \mathbb{R}$ denote the $i$-th Hermite polynomial. Note that $q_i$'s form an orthogonal basis for the Hilbert space of functions.:

$$\mathcal{H} = \left\{u : \mathbb{R} \to \mathbb{R} \mid \int u^2(x)\exp\left(-\frac{x^2}{2}\right) < \infty\right\},$$

which is equipped with the inner product

$$\langle u, v\rangle_{\mathcal{H}} = \frac{1}{\sqrt{2\pi}}\int u(x)v(x)\exp\left(-\frac{x^2}{2}\right)\mathrm{d}x$$

for $u, v \in \mathcal{H}$. We consider probabilist's convention of Hermite polynomial. Specifically, for $i, j \geq 0$, we have

$$\langle q_i, q_j\rangle_{\mathcal{H}} = \begin{cases} i! & i = j, \\ 0 & i \neq j. \end{cases} \tag{A.27}$$

Using the above orthogonal basis to decompose $\phi(W^0X)$, we have

$$\phi(W^0X) = \sum_{i=0}^{\infty} \frac{c_i}{i!} \cdot q_i(W^0X) \tag{A.28}$$

where $c_i = \langle\phi, q_i\rangle_{\mathcal{H}}$ and each matrix $q_i(W_0X) \in \mathbb{R}^{d_1 \times m}$ is formed by applying $q_i$ entry-wise to the matrix $W^0X$. Let us denote

$$M^0 := \phi(X^\top W^{0\top})\phi(W^0X).$$

In the following, we first obtain $\mathbb{E}[\tilde{M}^0] = \mathbb{E}[\phi(X^\top\tilde{W}^{0\top})\phi(\tilde{W}^0X)]$ with $\tilde{W}^0 \sim \mathcal{N}(0, 1)$ and then obtain a lower bound on $\sigma_{\min}(\mathbb{E}[M^0])$ and an upper bound on $\sigma_{\min}(\mathbb{E}[M^0])$ by appropriately scaling the data matrix $X$.

Applying Hermite decomposition (A.28) and taking expectation, we have

$$\mathbb{E}[\tilde{M}^0] = \mathbb{E}\left[\phi(X^\top\tilde{W}^{0\top})\phi(\tilde{W}^0X)\right]$$

$$= \sum_{i,j=0}^{\infty} \frac{c_ic_j}{i!j!}\mathbb{E}[q_i(X^\top\tilde{W}^{0\top})q_j(\tilde{W}^0X)] \tag{A.29}$$

where the expectation is w.r.t. the random matrix $\tilde{W}_0$. Let $\mathbf{x}_a \in \mathbb{R}^{d_0}$ denote the $a$-th column of the training data $X$. Each summand in (A.29) is an $m \times m$ matrix where

$$\left[\mathbb{E}[q_i(X^\top \tilde{W}^{0\top})q_j(\tilde{W}^0 X)]\right]_{a,b} = \sum_{c=1}^{d_1} \mathbb{E}\left[q_i(\mathbf{x}_a^\top \tilde{W}_{0,c,\rightarrow})q_j(\tilde{W}_{0,c,\rightarrow}^\top \mathbf{x}_b)\right], \tag{A.30}$$

where $\tilde{W}_{0,c,\rightarrow}$ is the $c$-th row of $\tilde{W}^0$ for $a, b \in [m]$.

In summand on the RHS of (A.30), we note that there is a linear combination of $\tilde{W}^0$'s elements inside of each Hermite polynomial.

We use the properties of Hermite polynomials (Olver et al., 2010)[§18.18.11]:

$$\frac{(a_1^2 + \cdots + a_r^2)^{\frac{i}{2}}}{i!}\tilde{q}_i\left(\frac{a_1 x_1 + \cdots + a_r x_r}{(a_1^2 + \cdots + a_r^2)^{\frac{1}{2}}}\right) = \sum_{s_1 + \cdots + s_r = i} \frac{a_1^{s_1} \cdots a_r^{s_r}}{s_1! \cdots s_r!}\tilde{q}_{s_1}(x_1) \cdots \tilde{q}_{s_r}(x_r) \tag{A.31}$$

where $\tilde{q}_i$'s form an orthogonal basis, equipped with the inner product $\langle u, v \rangle_{\tilde{\mathcal{H}}} = \frac{1}{\sqrt{\pi}}\int u(x)v(x)\exp(-x^2)\,\mathrm{d}x$. This basis follows the physicist's convention of Hermite polynomial.

Since $\tilde{q}_i$ and $q_i$ are rescalings of the other, we can replace $q_i$'s into (A.31). Note that we have $\|\mathbf{x}_a\|_2 = 1$ for all $a \in [n]$. Then we have

$$q_i(\mathbf{x}_a^\top \tilde{W}_{0,c,\rightarrow}) = i! \sum_{s_1 + \cdots + s_{d_0} = i} \frac{x_{a,1}^{s_1} \cdots x_{a,d_0}^{s_{d_0}}}{s_1! \cdots s_{d_0}!} q_{s_1}(\tilde{W}_{0,c,1}) \cdots q_{s_{d_0}}(\tilde{W}_{0,c,d_0}) \tag{A.32}$$

where $x_{a,k}$ and $\tilde{W}_{0,c,k}$ are $k$-th entry of $\mathbf{x}_a$ and $\tilde{W}_{0,c,\rightarrow}$ for $k \in [d_0]$. Using the expansion in (A.32), we expand (A.30) as follows:

$$\begin{aligned}
\zeta_{i,j}(a,b) &= i!j! \sum_{s_1 + \cdots + s_{d_0} = i} \sum_{s_1' + \cdots + s_{d_0}' = j} \frac{x_{a,1}^{s_1} \cdots x_{a,d_0}^{s_{d_0}}}{s_1! \cdots s_{d_0}!} \cdot \frac{x_{b,1}^{s_1'} \cdots x_{b,d_0}^{s_{d_0}'}}{s_1'! \cdots s_{d_0}'!} \rho_{\mathbf{s},\mathbf{s}'}(\tilde{W}_{0,c,\rightarrow}) \\
&= \begin{cases} (i!)^2 \sum_{s_1 + \cdots + s_{d_0} = i} \frac{(x_{a,1}x_{b,1})^{s_1} \cdots (x_{a,d_0}x_{b,d_0})^{s_{d_0}}}{s_1! \cdots s_{d_0}!} & i = j, \\ 0 & i \neq j \end{cases} \\
&= \begin{cases} i! \sum_{s_1 + \cdots + s_{d_0} = i} \binom{i}{s_1, \cdots, s_{d_0}}(x_{a,1}x_{b,1})^{s_1} \cdots (x_{a,d_0}x_{b,d_0})^{s_{d_0}} & i = j, \\ 0 & i \neq j \end{cases}
\end{aligned} \tag{A.33}$$

where $\zeta_{i,j}(a,b) = \mathbb{E}\left[q_i(\mathbf{x}_a^\top \tilde{W}_{0,c,\rightarrow})q_j(\tilde{W}_{0,c,\rightarrow}^\top \mathbf{x}_b)\right]$,

$$\rho_{\mathbf{s},\mathbf{s}'}(\tilde{W}_{0,c,\rightarrow}) = \mathbb{E}\left[q_{s_1}(\tilde{W}_{0,c,1}) \cdots q_{s_{d_0}}(\tilde{W}_{0,c,d_0}) \cdot q_{s_1'}(\tilde{W}_{0,c,1}) \cdots q_{s_{d_0}'}(\tilde{W}_{0,c,d_0})\right],$$

$\mathbf{s} = [s_1, \cdots, s_{d_0}]$, and $\mathbf{s}' = [s_1', \cdots, s_{d_0}']$.

To simplify the expression in (A.33), we define $X^{*i} \in \mathbb{R}^{d_0^i \times m}$ where the $a$-th column is given by

$$X_a^{*i} = \mathrm{vec}(\mathbf{x}_a \otimes \cdots \otimes \mathbf{x}_a) \in \mathbb{R}^{d_0^i},$$

which is also called Khatri-Rao product. For $i = 0$, we use the convention that $X^{*0} = \mathbf{1}\mathbf{1}^\top \in \mathbb{R}^{m \times m}$.

We can rewrite (A.33) as follows:

$$\zeta_{i,j}(a,b) = \begin{cases} i!\langle X_a^{*i}, X_b^{*i} \rangle & i = j \\ 0 & i \neq j. \end{cases} \tag{A.34}$$

Substituting (A.34) back into (A.30), we find that

$$\begin{aligned}
\left[\mathbb{E}[q_i(X^\top \tilde{W}^{0\top})q_j(\tilde{W}^0 X)]\right]_{a,b} &= \sum_{c=1}^{d_1} \mathbb{E}\left[q_i(\mathbf{x}_a^\top \tilde{W}_{0,c,\rightarrow})q_j(\tilde{W}_{0,c,\rightarrow}^\top \mathbf{x}_b)\right] \\
&= \begin{cases} d_1 i!\langle X_a^{*i}, X_b^{*i} \rangle & i = j \\ 0 & i \neq j. \end{cases}
\end{aligned} \tag{A.35}$$

Substituting (A.35) into (A.29), we have

$$\mathbb{E}\left[\tilde{M}^0\right] = d_1 \left( c_0^2 \mathbf{1}\mathbf{1}^\top + c_1^2 X^\top X + \sum_{i=2}^\infty \frac{c_i^2}{i!} (X^{*i})^\top X^{*i} \right). \tag{A.36}$$

We now establish an upper bound on $\sigma_{\max}\left( \sum_{i=2}^\infty \frac{c_i^2}{i!} (X^{*i})^\top X^{*i} \right)$:

$$\sigma_{\max}\left( \sum_{i=2}^\infty \frac{c_i^2}{i!} (X^{*i})^\top X^{*i} \right) \le \sum_{i=2}^\infty \frac{c_i^2}{i!} \sigma_{\max}((X^{*i})^\top X^{*i}) \tag{A.37}$$
$$\le c_\infty^2 \sigma_{\max}^2(X)$$

where $c_\infty$ is given by

$$c_\infty^2 = \sum_{i=2}^\infty \frac{c_i^2}{i!},$$

which is finite provided that $\|\phi\|_{\mathcal{H}}$ is bounded.

Using (A.37), we now establish an upper bound on $\sigma_{\max}(\mathbb{E}[\tilde{M}^0])$:

$$\sigma_{\max}(\mathbb{E}[\tilde{M}^0]) \lesssim d_1 \left( m c_0^2 + (c_1^2 + c_\infty^2)\sigma_{\max}^2(X) \right)$$

Moreover, suppose there exists some $t$ such that $\sigma_{\min}(X^{*t}) > 0$. Then we have $d_1 \frac{c_t^2}{t!} \sigma_{\min}^2(X^{*t}) \lesssim \sigma_{\min}(\mathbb{E}[M^0])$. This requires to have $d_0^t \ge m$. Putting together the lower bound on $\sigma_{\min}(\mathbb{E}[\tilde{M}^0])$ and the upper bound on $\sigma_{\min}(\mathbb{E}[\tilde{M}^0])$, noting $W^0 = \omega_1 \tilde{W}^0$, and scaling $X$ accordingly to take into account the coefficient $\omega_1$, we have

$$\omega_1^{2t} d_1 \frac{c_t^2}{t!} \sigma_{\min}^2(X^{*t}) \lesssim \sigma_{\min}(\mathbb{E}[M^0]) \le \sigma_{\max}(\mathbb{E}[M^0]) \lesssim d_1 \left( m c_0^2 + \omega_1^2(c_1^2 + c_\infty^2)\sigma_{\max}^2(X) \right). \tag{A.38}$$

### E.2 Concentration of the random matrix $M^0$

To see how well the random matrix $M^0$ concentrates about its expectation, note that

$$M^0 = \phi(X^\top W^{0\top})\phi(W^0 X)$$
$$= \sum_{i=1}^{d_1} \phi(X^\top W_{0,i,\to}^\top)\phi(W_{0,i,\to} X) \tag{A.39}$$
$$= \sum_{i=1}^{d_1} A_i$$

where $\{A_i\}_{i=1}^{d_1} \subset \mathbb{R}^{m \times m}$ are independent random matrices.

Consider the event $\mathcal{E}_1$ that

$$\max_{i \in [d_1]} \|W_{0,i,\to}\|_2 \lesssim k_1 \omega_1 \sqrt{d_0 \log d_1}, \quad \max_{i \in [d_1]} \|V_{0,i,\downarrow}\|_2 \lesssim k_2 \omega_2 \sqrt{d_2 \log d_1} \tag{A.40}$$

where $V_{0,i,\downarrow}$ is the $i$-th column of $V_0$. Note that $W_{0,i,\to} \in \mathbb{R}^{d_0}$ and $V_{0,i,\downarrow} \in \mathbb{R}^{d_2}$ are random zero-mean Gaussian vectors whose entries' variances are $\omega_1^2$ and $\omega_2^2$, respectively. Therefore, with an application of the scalar Bernstein inequality (Vershynin, 2012, Proposition 5.16), followed by the union bound, we observe that the event $\mathcal{E}_1$ happens except with a probability of at most

$$p_1 := d_1^{-Ck_1 d_0} + d_1^{-Ck_2 d_2}, \tag{A.41}$$

for a universal constant $C$ with sufficiently large $k_1, k_2$.

Let $i \in [d_1]$. Conditioned on the event $\mathcal{E}_1$, an upper bound on $\|\phi(X^\top W_{0,i,\to})\|_2$ is given by:

$$\|\phi(X^\top W_{0,i,\to})\|_2 \lesssim \dot{\phi}_{\max}\sigma_{\max}(X)k_1\omega_1\sqrt{d_0 \log d_1}. \tag{A.42}$$

Moreover, we have

$$\begin{aligned}
\sigma_{\max}(A_i) &= \|\phi(X^\top W_{0,i,\to})\|_2^2 \\
&= \|\phi(X^\top W_{0,i,\to}) - \phi(0)\|_2^2 \\
&\lesssim \dot{\phi}_{\max}^2\sigma_{\max}^2(X)k_1^2\omega_1^2 d_0 \log d_1.
\end{aligned} \tag{A.43}$$

We now focus on the concentration of $\sigma_{\min}(M^0)$ and $\sigma_{\max}(M^0)$. We use a concentration property, which provides the tail bound of $\tilde{f}(W) = \phi(X^\top W^\top)\phi(WX)$ with multivariate Gaussian input $W$. In the following lemma, we show that $\tilde{f}$ is a Lipschitz function, and its Lipschitz constant explains how $\tilde{f}(W)$ concentrates around its mean.

**Lemma A.7.** *Let $\tilde{f}(W) = \phi(X^\top W^\top)\phi(WX)$. Suppose $W$ satisfies (A.40). Then $\tilde{f}$ is $\kappa$-Lipschitz function with constant $\kappa = 4\dot{\phi}_{\max}^2\sigma_{\max}^2(X)k_1\omega_1\sqrt{d_0 \log d_1}$. So we have*

$$\|\tilde{f}(W) - \tilde{f}(W')\| < 4\dot{\phi}_{\max}^2\sigma_{\max}^2(X)k_1\omega_1\sqrt{d_0 \log d_1} \cdot \|W - W'\|.$$

*Proof.* Note that $\tilde{f}(W^0) = M^0$ and $\tilde{f}$ can be represented as

$$\tilde{f}(X) = \sum_{i=1}^{d_1} f_i(W_{i,\to})$$

where $f_i$ is given by $f_i(W_{i,\to}) = \phi(X^\top W_{i,\to}^\top)\phi(W_{i,\to}X)$. We prove that each $f_i$ is $\kappa$-Lipschitz, which implies that $\tilde{f}$ is also $\kappa$-Lipschitz.

We note that $f_i$'s can be expressed as a composition of three functions:

$$f_i(\mathbf{v}) = (g_1 \circ g_2 \circ g_3)(\mathbf{v})$$

where $g_1$, $g_2$, and $g_3$ are given by

$$g_1(\mathbf{v}) = \mathbf{v}\mathbf{v}^\top, \; f_2(\mathbf{v}) = \phi(\mathbf{v}), \; f_3(\mathbf{v}) = \mathbf{v}X. \tag{A.44}$$

It is clear that $g_2$ is $\dot{\phi}_{\max}$-Lipschitz, and $g_3$ is $\sigma_{\max}(X)$-Lipschitz from their definitions. Lipschitz constant of $g_1$ comes from the domain bound as follows:

$$\begin{aligned}
\|g_1(\mathbf{v} + \delta\mathbf{v}) - g_1(\mathbf{v})\| &= \|\delta\mathbf{v}\mathbf{v}^\top + \mathbf{v}\delta\mathbf{v}^\top + \delta\mathbf{v}\delta\mathbf{v}^\top\| \\
&\leq 2\|\delta\mathbf{v}\mathbf{v}^\top\| + \|\delta\mathbf{v}\delta\mathbf{v}^\top\| \\
&\leq (2\|\mathbf{v}\| + \|\delta\mathbf{v}\|) \cdot \|\delta\mathbf{v}\|.
\end{aligned} \tag{A.45}$$

A bound on $(2\|\mathbf{v}\| + \|\delta\mathbf{v}\|)$ is obtained in (A.42). Then $g_1$ is $\kappa_1$-Lipschitz function with $\kappa_1 = 4\dot{\phi}_{\max}\sigma_{\max}(X)k_1\omega_1\sqrt{d_0 \log d_1}$. Therefore, all $g_1$, $g_2$ and $g_3$ are Lipschitz function, so their composition $f_i$ is also Lipschitz function with constant $\kappa = 4\dot{\phi}_{\max}^2\sigma_{\max}^2(X)k_1\omega_1\sqrt{d_0 \log d_1}$, which completes the proof. □

**Lemma A.8.** *Let $\mathbf{z} \in \mathbb{R}^d$ denote a Gaussian random vector. Then we have $\Pr\{\|\mathbf{z} - \mathbb{E}[\mathbf{z}]\| > t \,|\mathcal{E}_2\} \lesssim \exp(-t^2)$ where $\mathcal{E}_2$ is the event that $\|\mathbf{z}\|$ is bounded.*

We can focus on the tail distribution of $M_0 = \tilde{f}(W_0)$. Using Lemmas A.7 and A.8, we have

$$\Pr\{\|M^0 - \mathbb{E}[M^0]\| > t \,|\mathcal{E}_1\} \lesssim \exp(-k_3^2) \tag{A.46}$$

where $t = k_3 4\dot{\phi}_{\max}^2\sigma_{\max}^2(X)k_1\omega_1\sqrt{d_0 \log d_1}$ with some constant $k_3$.

Using (A.46), we now establish a tail bound on $\sigma_{\min}(M^0)$:

$$
\begin{aligned}
\Pr\{\sigma_{\min}(M^0) \le (1-\delta_1)\sigma_{\min}(\mathbb{E}[M^0])|\mathcal{E}_1\} &\le \Pr\{|\sigma_{\min}(M^0) - \sigma_{\min}(\mathbb{E}[M^0])| \ge \delta_1\sigma_{\min}(\mathbb{E}[M^0])|\mathcal{E}_1\} \\
&\le \Pr\{\sigma_{\min}(M^0 - \mathbb{E}[M^0]) \ge \delta_1\sigma_{\min}(\mathbb{E}[M^0])|\mathcal{E}_1\} \\
&\le \Pr\{\sigma_{\max}(M^0 - \mathbb{E}[M^0]) \ge \delta_1\sigma_{\min}(\mathbb{E}[M^0])|\mathcal{E}_1\} \\
&\le \Pr\{\|M^0 - \mathbb{E}[M^0]\| \ge \delta_1\sigma_{\min}(\mathbb{E}[M^0])|\mathcal{E}_1\} \\
&\lesssim p_2
\end{aligned}
$$

where

$$
p_2 = \exp\left(-\left(\frac{\delta_1\sigma_{\min}(\mathbb{E}[M^0])}{4\dot{\phi}_{\max}^2\sigma_{\max}^2(X)k_1\omega_1\sqrt{d_0\log d_1}}\right)^2\right).
$$

Similarly, we obtain

$$
\Pr\{\sigma_{\max}(M^0) \ge (1+\delta_2)\sigma_{\max}(\mathbb{E}[M^0])|\mathcal{E}_1\} \lesssim p_3
$$

where

$$
p_3 = \exp\left(-\left(\frac{\delta_2\sigma_{\max}(\mathbb{E}[M^0])}{4\dot{\phi}_{\max}^2\sigma_{\max}^2(X)k_1\omega_1\sqrt{d_0\log d_1}}\right)^2\right).
$$

Putting these bounds together with (A.38), we have :

$$
\begin{aligned}
\omega_1^t\sqrt{(1-\delta_1)\frac{c_t^2}{t!}d_1}\sigma_{\min}(X^{*t}) &\le \sigma_{\min}(\phi(W^0X)) \\
\sigma_{\max}(\phi(W^0X)) &\le \sqrt{(1+\delta_2)}(\omega_1\sqrt{(c_1^2+c_\infty^2)d_1}\sigma_{\max}(X) + |c_0|\sqrt{d_1m})
\end{aligned}
\tag{A.47}
$$

except with a probability of at most $p_1 + p_2 + p_3$.

With establishing the bounds on $\sigma_{\min}(\phi(W^0X))$ and $\sigma_{\max}(\phi(W^0X))$, we can finally estimate $\mu_\Phi, \nu_\Phi$ as follows:

### E.3 LOWER BOUND ON $\mu_\Phi$

A lower bound on $\mu_\Phi$ is given by

$$
\omega_1^t\sqrt{(1-\delta_1)\frac{c_t^2}{t!}d_1}\sigma_{\min}(X^{*t}) \le \sigma_{\min}(\phi(W^0X)) = \mu_\Phi,
\tag{A.48}
$$

except with a probability of at most $p_1 + p_2$.

### E.4 UPPER BOUND ON $\nu_\Phi$

Since $\nu_\Phi = \dot{\phi}_{\max}\sigma_{\max}(X)\sigma_{\max}(V^0) + \sigma_{\max}(\phi(W^0X))$, we obtain a bound on $\sigma_{\max}(V^0)$:

Since $V^0$ is a Gaussian random matrix, we have

$$
\sigma_{\max}(V^0) \le \omega_2(2\sqrt{d_1} + \sqrt{d_2}) \lesssim \omega_2\sqrt{d_1}
\tag{A.49}
$$

except with a probability of at most $p_4 = \exp(-Cd_1)$ where $C$ is a universal constant (Vershynin, 2012)[Corollary 5.35].

Combining (A.49) with the upper bound on $\sigma_{\max}(\phi(W^0X))$, we have

$$
\begin{aligned}
\nu_\Phi &= \dot{\phi}_{\max}\sigma_{\max}(X)\sigma_{\max}(V^0) + \sigma_{\max}(\phi(W_0X)) \\
&\lesssim \omega_2\dot{\phi}_{\max}\sigma_{\max}(X)\sqrt{d_1} + \omega_1\sqrt{(1+\delta_2)(c_1^2+c_\infty^2)d_1}\sigma_{\max}(X) + c_0\sqrt{(1+\delta_2)d_1m}
\end{aligned}
$$

except with a probability of at most $p_1 + p_3 + p_4$.

### E.5 UPPER BOUND ON $h(\Theta^0)$

In this section, we bound $h(\Theta^0)$. Using $\|\mathbf{a} + \mathbf{b}\|_2^2 \leq 2\|\mathbf{a}\|_2^2 + 2\|\mathbf{b}\|_2^2$, we have

$$
\begin{aligned}
h(\Theta^0) &= \frac{1}{m}\|V^0\phi(W^0X) - Y\|^2 \\
&\leq \frac{2}{m}\|V^0\phi(W^0X)\|^2 + \frac{2}{m}\|Y\|^2.
\end{aligned}
\tag{A.50}
$$

To upper bound the random norm in (A.50), we first decompose $V^0\phi(W^0X)$ into terms including $W_{0,i,\to} \in \mathbb{R}^{d_0}$ and $V_{0,i,\downarrow} \in \mathbb{R}^{d_2}$ as follows:

$$
V^0\phi(W^0X) = \sum_{i=1}^{d_1} B_i
\tag{A.51}
$$

where $B_i = V_{0,i,\downarrow}\phi(W_{0,i,\to}^\top X) \in \mathbb{R}^{d_2 \times m}$'s are independent random matrices for $i \in [d_1]$.

Conditioned on the event $\mathcal{E}_1$ defined in (A.40), we bound $\|B_i\|$:

$$
\begin{aligned}
\|B_i\| &= \|V_{0,i,\downarrow}\|_2\|\phi(W_{0,i,\to}^\top X)\|_2 \\
&\leq \|V_{0,i,\downarrow}\|_2 \cdot \dot{\phi}_{\max}\sigma_{\max}(X)k_1\omega_1\sqrt{d_0\log d_1} \\
&\leq \omega_1\omega_2\dot{\phi}_{\max}\sigma_{\max}(X)k_1k_2\sqrt{d_0d_2}\log d_1
\end{aligned}
\tag{A.52}
$$

for $i \leq d_1$.

Substituting the upper bound in A.51 into A.52 and applying the Hoeffding inequality (Hoeffding, 1963), we have

$$
\begin{aligned}
\Pr\{\|V^0\phi(W^0X)\| \gtrsim u(d_0, d_1, d_2)|\mathcal{E}_1\} &= \Pr\{\|V^0\phi(W^0X) - \mathbb{E}[V^0\phi(W^0X))|\mathcal{E}_1]\| \gtrsim u(d_0, d_1, d_2)|\mathcal{E}_1\} \\
&\leq \Pr\left\{\sum_{i=1}^{d_1}\|B_i - \mathbb{E}[B_i]\| \gtrsim u(d_0, d_1, d_2)|\mathcal{E}_1\right\} \\
&\leq p_5
\end{aligned}
$$

where

$$
u(d_0, d_1, d_2) = \delta_3\omega_1\omega_2\dot{\phi}_{\max}k_1k_2\sqrt{d_0d_1d_2}\sigma_{\max}(X)\log d_1
$$

and $p_5 = \exp(-C\delta_3^2)$ with $\delta_3 \geq 0$ and a universal constant $C$.

Therefore, under the event $\mathcal{E}_1$, we have

$$
\begin{aligned}
h(\Theta^0) &\leq \frac{2}{m}\|V^0\phi(W^0X)\|^2 + \frac{2}{m}\|Y\|^2 \\
&\lesssim \frac{1}{m}\delta_3^2\omega_1^2\omega_2^2\dot{\phi}_{\max}^2k_1^2k_2^2d_0d_1d_2\sigma_{\max}^2(X)\log^2 d_1 + \frac{1}{m}\|Y\|^2
\end{aligned}
\tag{A.53}
$$

except with a probability of at most $p_1 + p_5$. It is natural to assume that $d_2 = o(d_1)$. We also have $\|Y\| \leq 1$.

Suppose that

$$
\omega_1\omega_2 \lesssim \frac{1}{\dot{\phi}_{\max}\sqrt{d_0d_1}\log d_1}.
\tag{A.54}
$$

Substituting (A.54) into (A.53), we have

$$
h(\Theta^0) \leq \frac{1}{m}\delta_3^2k_1^2k_2^2\sigma_{\max}^2(X)
\tag{A.55}
$$

where $\delta_3$, $k_1$, and $k_2$ are all constants and independent of $d_0$, $d_1$, and $m$.

