# OpenReview forum: "Linear Convergence of SGD on Overparametrized Shallow Neural Networks"
_ICLR.cc/2022/Conference — ICLR 2022 Submitted_

### Official Review · Reviewer_2P9T · 2021-10-23

**Correctness:** 4
**Technical Novelty And Significance:** 2
**Empirical Novelty And Significance:** 1
**Recommendation:** 5
**Confidence:** 4

**Main Review:**

Pros:
1. This paper relaxed the over-parameterization requirement in two-layer neural networks for the SGD algorithm, which is certainly an important direction.
2. The proof involves bounding the distance to initialization for SGD iterates, which is much more challenging than for GD. I think the techniques here can be useful in future studies in SGD convergence.

Cons:
1. I am concerned about the novelty of the proof. It seems to me that the proof is very similar to the NTK analysis in the sense that it argues the Jacobian matrix is full-rank during the training because the SGD iterates stay close to the initialization. I wonder if there is any conceptual difference between this proof and the NTK argument.
2. In section 3.1, the authors assumed that $m\geq d_0^2$ and claimed it's the case in practice. I think in practice the number of training samples $(m)$ is much smaller than the square of input dimension $(d_0^2).$ For example, in CIFAR the input dimension is $32\times 32\times 3=3072$ and there are only $50000$ training samples; in ImageNet the input dimension is $256\times 256\times 3=196608$ and there are only 1M samples.
3. One of the messages of this paper is that a smaller batch size requires a wider neural network. Is there any empirical experiment that can support this claim? Otherwise, this might just be an artifact of the analysis.

**Summary Of The Paper:**

1. This paper proves that SGD converges to a global minimum in certain non-convex problems assuming the loss function satisfies a growth condition. The proof relies on assuming that the initial Jacobian matrix is non-singular and shows that it stays non-singular since SGD iterates remain close to the initialization.
2. The paper then applies the above analysis to a two-layer neural network and proves that a subquadratic scaling on the width is sufficient for global convergence assuming the minibatch size grows with the number of training samples. For constant batch size, it requires quadratic over-parameterization. Furthermore, an interpolation between subquadratic and quadratic scaling is given depending on the batch size.

**Summary Of The Review:**

The main result of this paper is proving the subquadratic width scaling for SGD convergence in two-layer neural networks assuming the batch size grows with the number of training samples. I understand that the analysis requires new techniques to bound the distance of SGD iterates to its initialization, but I think the proof is conceptually the same as NTK analysis and am concerned about its novelty. Therefore, I think this paper is marginally below the acceptance threshold. I will be willing to raise my score if my concerns are addressed.

------------------

Thanks for the response. After reading the response and other reviews, I decided to keep my original score. My major concern is still the novelty of the proof.

---

> ### Author Response · Authors · 2021-11-19
> **Reply to Reviewer 2P9T**
>
> We thank the reviewer for their thoughtful comments which we address one by one below.
>
> > **Q1.** Is there any conceptual difference between this proof and the NTK argument?
>
> Our proof is based on showing that under some proper initialization and learning rate, we can control the expected length of the trajectory, regardless of the number of iterations. To do so, we use smoothness of the mapping $\Phi$ and show that we can indeed bound the smoothness constant of $\Phi$ restricted to a neighbourhood of the initialization. Our proof also highlights the role of the strong growth condition on the convergence of SGD with high probability, which is not the case in the NTK argument.
>
> ________
> > **Q2.** "In section 3.1, the authors assumed that $m≥d_0^2$ ... "
>
> Concerning the bound on the number of features in Section 3.1, please note that $d_0$ is in Big omega of $\sqrt{m}$, i.e., $m\lesssim d_0^2$, which is the case for CIFAR10 and ImageNet.
>
> ________
> > **Q3.** Experimental studies supporting smaller batch size requires a wider neural network
>
> Extensive experimental results are provided in (Park et al., 2019) where the authors show that the optimal batch size is inversely proportional to the widening factor.
>
> Daniel Park, Jascha Sohl-Dickstein, Quoc Le, and Samuel Smith. The effect of network width on stochastic gradient descent and generalization: an empirical study. In International Conference on Machine Learning (ICML), 2019.

---

### Official Review · Reviewer_t48E · 2021-10-24

**Correctness:** 4
**Technical Novelty And Significance:** 2
**Empirical Novelty And Significance:** 2
**Recommendation:** 3
**Confidence:** 3

**Main Review:**

-Strengths:

1. This paper provided a better width requirement bound for linear convergence in two-layer neural network training by generalizing techniques from previous results and taking batch size into consideration. Specifically, compared to (Oymak & Soltanolkotabi, 2020), this paper enables the training for both layers instead of a single layer. Besides, compared to (Anonymous), this paper enables training using mini-batch SGD instead of GD.

2. This paper is generally well-organized and easy to follow. It also provided detailed proof sketches with a few remarks, making the proof easy to understand.

3. The related works appear to be adequately cited and compared in detail.

-Concerns:

1. The novelty of this paper might be somewhat limited, and this is my major concern of this paper. I will explain in detail in the following paragraphs.

1.1 The learning dynamics of neural networks in this paper are still in the lazy regime, and the proof ideas are of similar styles as previous papers in this area. Specifically, as shown in Lemma 1, the authors require that the weights do not move far (i.e., larger than a constant) from the initialization to ensure the local smoothness and bound the eigenvalues of the Jacobian. The bound on minimum eigenvalue of the Jacobian ensures linear convergence, which results in bounded weight change and then induces bounds on eigenvalues of the Jacobian. Therefore, the learning dynamics in this paper still require a lot of things (including weights, Jacobian matrices, etc.) to remain close to initialization, which should be considered as lazy regime. This is different from the neural network training in practice since empirically the weights of neural networks usually move far from initialization.

1.2 The novelty of the techniques used in this paper seems a bit limited. The bounds for local smoothness $\beta_\Phi$, eigenvalues of $\Phi^*(w_0)$, initial function value, and preservation of "near-isometry" during training all come from (Anonymous). Furthermore, the idea of bounding the probability that the weights leave the local neighborhood of the initialization exists in Theorem 3.1 of (Oymak & Soltanolkotabi, 2019) and Theorem 2.6 of (Oymak & Soltanolkotabi, 2020). This paper provided a new way of bounding the expected length of SGD trajectory (Proposition 1), enabled training for the second-layer weights, and generalized SGD to mini-batch SGD by introducing the batch size $b$ (Theorem 2) to interpolate between (Anonymous) and (Oymak & Soltanolkotabi, 2020). The techniques used in these steps seem specific to the particular setting in this paper and might not generalize. Thus, the technical novelty of this paper might be a bit limited.

2. The results in this paper only apply to smooth activation functions with bounded Hermite norm, which is a bit strong, and the width requirement also depends on the smoothness factor and Hermite norm. Most previous works hold for ReLU activations, as also shown in Table 1.

-Minor Comments:

1. The statement of Theorem 1 might need to be further clarified about the randomness. When stating Theorem 1, the authors said "with probability at least $1-\zeta$", and the result still has the expectation notation. It would be better if the authors could clarify what randomness goes into the "$1-\zeta$" part and what randomness is the expectation taken over. One possible way might be to define the event that SGD travels outside the neighborhood of the initialization and state Theorem 1 in a similar way as Theorem 2.6 in (Oymak & Soltanolkotabi, 2020).

-Typos:

1. Definition 3, "$\frac1c$" -> ""$\frac1b$""
2. $\rho_\Phi$ is used in Proposition 1 but not formally defined until Lemma 3, so it is better to move the definition earlier.
3. "$\simeq$" is used in Lemma 2 and Section 3.1 but is not formally defined.

---------------------Update--------------------------

Thank the authors for their responses. I have read all other reviews and the authors' responses to all the reviewers, and I have decided to keep my score unchanged. The authors' responses address some of my concerns, e.g., their main results do not directly require the weights during training remain close enough to the initialization. However, my major concern (the novelty of the proof techniques, which is also shared by the other reviewers) still remains because most of the results/methods used in this paper already exist in the literature. Therefore, I would like to keep my score and tend to recommend rejection.

**Summary Of The Paper:**

This paper proved that a two-layer neural network with smooth activation and proper initialization can converge linearly to a global minima of training loss using mini-batch SGD when the width is larger than $\Omega(m^2/\sqrt{b})$ where $m$ is the number of training data and $b$ is batch size. As the batch size increases, this provides an interpolation between the quadratic result for SGD and the sub-quadratic result for full-batch GD. To prove this result, the authors first provide a general convergence result for SGD on a particular class of functions and then apply this framework to 2-layer neural networks to derive the width requirement.

**Summary Of The Review:**

I tend to vote for rejecting this paper. Despite generalizing the linear convergence result to mini-batch SGD and improving the width requirement to sub-quadratic, my major concern about this paper is its limited novelty, as explained in the "Concerns" section above.

---

> ### Author Response · Authors · 2021-11-19
> **Reply to Reviewer t48E**
>
> We thank the reviewer for their thoughtful comments which we address one by one below.
>
> > **Q1.**  the learning dynamics in this paper still require a lot of things (including weights, Jacobian matrices, etc.) to remain close to initialization, which should be considered as lazy regime.
>
> We emphasize that we do not assume that the weights remain close to the initialization. We instead show in Proposition 1 that they remain in a bounded neighborhood of the initialization, i.e., that they do not diverge, which is a necessary condition for convergence of the method. The radius of the ball can actually be quite big, depending on the smoothness and near-isometry constants at the initialization. Hence, we cannot conclude that this leads to lazy training.
>
> The only assumptions that we use in our final result (Theorem 2) are Assumptions 3 and 4 (in addition to a data assumption on the minimum singular value of the data matrix, as pointed out by reviewer Vg9p).
>
> Lemma 1 has nothing to do with the dynamics of SGD, but with the smoothness and near isometry constants of the neural network mapping. We believe you are referring to Proposition 1.
> ________
> > **Q2.**  The results in this paper only apply to smooth activation functions with bounded Hermite norm, which is a bit strong, and the width requirement also depends on the smoothness factor and Hermite norm. Most previous works hold for ReLU activations, as also shown in Table 1.
>
>
> Regarding ReLU, we would like to emphasize that our assumptions hold for smooth approximations of ReLU such as GeLU and softplus, which often achieve similar performance in practice (see discussion after Assumption 3). However, to extend to ReLU we would have to deal with isolated non-differentiable points. One potential idea to extend our results to ReLU networks is to define $\nabla\Phi$ using generalized gradients [Frank Clarke, 1975].
>
> We argue that the differentiability assumption  is not a drawback diminishing the significance of our results:
> 1) Differentiable activation functions are widely used in practice.
> 2) The differentiability assumption is also used in other overparameterization works (see ref [Liu et al., 2020]).
> 3) The main goal in the literature is to understand when optimization is successful for a neural network of increasing width. Given that the goal is to understand the general behaviour, showing results for differentiable activation functions would translate to our  understanding of ReLU as it can be seen as a limit case if we  consider a sequence of smooth approximations of ReLU.
> 4) Focusing on a particular fixed activation function can be seen as limited, compared to using differentiability assumptions that encompass many different activations used in practice, as well as arbitrary smooth approximations of ReLU.
> 5) Using ReLU makes the whole objective non-smooth which makes it much harder to rigorously talk about the gradient of the function, as we remark in the introduction.
>
>
> The bounded Hermite norm is a mild assumption, which holds for GeLU, sigmoid, and tanh.
>
>
>
> Chaoyue Liu, Libin Zhu, and Mikhail Belkin. Loss landscapes and optimization in over- parameterized non-linear systems and neural networks. In Advances in neural information processing systems (NeurIPS), 2020.
>
> Frank H. Clarke. Generalized gradients and applications. Transactions of the American Mathematical Society, 205:247–247, 1975.

---

### Official Review · Reviewer_SWMd · 2021-11-01

**Correctness:** 2
**Technical Novelty And Significance:** 1
**Empirical Novelty And Significance:** Not applicable
**Recommendation:** 1
**Confidence:** 5

**Details Of Ethics Concerns:**

1: The results of the paper highly overlap with a publicly accessable work (Liu, et al. Loss landscapes and optimization in over-parameterized non-linear systems and neural networks, 2021). Just raise this point to see whether there is any plagiarism.


**Main Review:**

The results presented in this paper are already known in the literature. Please check the two highly related papers: Liu et al (2021), Liu et al (2020).

>Specifically, assuming PL condition on the loss function, convergence of SGD is already obtained in Liu et al (2021), with an exponential learning rate. This covers Theorem 1 of this submission.

>The concepts of near-isometry (definition 4) and length of trajectory (proposition 1) are also similarly established in Liu et al (2021). Near-isometry corresponds to the so-called uniform conditioning of NTK (neural tangent kernel), and proposition 1 corresponds to the “existence of solution” in a neighborhood ball.

>PL conditions are shown on neural networks in Liu et al (2021). I noticed that Liu et al (2021) includes deep neural networks, as well as shallow ones. But this submission only considers shallow networks, with more assumptions (assumption 3 & 4).

The proof of the growth condition of the shallow network is incorrect. In section 2, the theory assumes growth condition on stochastic gradients, i.e., derivative of loss w.r.t. the model *parameters*. However, when proving a shallow network satisfies growth conditions in the proof of theorem 2, the paper analyzes a different object:  derivative of loss w.r.t. the model *output*.

The notations of the paper are not consistent and ambiguous, which makes it hard to read. For example, in Eq.(13) and the sentence after, matrix Z is discussed, but Z is never defined before.


References:

Liu, et al. Loss landscapes and optimization in over-parameterized non-linear systems and neural networks, 2021.

Liu, et al. On the linearity of large non-linear models: when and why the tangent kernel is constant. NeurIPS, 2020.

================ Edited after author feedback ================

Thanks for the authors for the feedback.

The paper seems properly cited the anonymous paper. So, I removed the dual submission concern flag in the ethic review part.

As for the high similarities with prior works, I never "acknowledged that there are *significant* differences between" the current submission and  prior works, e.g., [Liu et al., 2020]. The concepts and main logic are highly similar to prior works I mentioned above.

I have read the other reviews and agree that this paper lacks novelty, and that main concepts and methods are already known in prior works. So, I would like to keep my score unchanged.





**Summary Of The Paper:**

This paper first analyzes the convergence of SGD under assumptions of PL condition on loss function and growth condition on stochastic gradients. Then, it considers a two-layer neural network with some special setting, and claims that this network with quadratic loss satisfies both PL condition and growth condition, and hence SGD converges.

**Summary Of The Review:**

1. Known results
2. incorrect proof

---

> ### Author Response · Authors · 2021-11-19
> **Reply to Reviewer SWMd**
>
> We thank the reviewer for their thoughtful comments which we address one by one below.
>
> ________
> > **Q1.** "The proof of the growth condition of the shallow network is incorrect ... "
>
> Concerning the proof of growth condition, we assume the loss function satisfies the growth condition when we take derivatives of $f$ w.r.t the model output. The loss function $f$ takes the model output as its input, not the parameters, which is specified in Eq. (3). It is different from the objective function $h$, which takes parameters as its input. As stated in Assumption 2, we assume that $f$ satisfies the growth condition, which holds for the special case of quadratic loss as shown in Theorem 2.
> ________
> > **Q2.** "...  in Eq.(13) and the sentence after, matrix Z is discussed, but Z is never defined before"
>
> Regarding notations, in the case of neural networks, the model output (input of function $f$) becomes a matrix. In Eq. (13), we wrote the expression for the quadratic loss function with some matrix $Z$ as its input. We did not introduce $Z$ earlier, because we first formulated the problem considering a general function $f$ in Eq. (3).

---

> ### Author Response · Authors · 2021-11-19
> **Reply to Reviewer SWMd concerning ethic flag**
>
>
> > The results presented in this paper are already known in the literature.
> ... assuming PL condition on the loss function, convergence of SGD is already obtained in Liu et al (2021)...
> ...Near-isometry corresponds to the so-called uniform conditioning of NTK and proposition 1 corresponds to the “existence of solution” in a neighborhood ball..
> ...  Liu et al (2021) includes deep neural networks, as well as shallow ones...
>
> We indeed missed [Liu, et al. Loss landscapes and optimization in over-parameterized non-linear systems and neural networks, 2021] as a related work and will make sure to cite it since there are similarities in terms of proof techniques. However, we strongly believe that there are significant differences between these papers in terms of assumptions, techniques, and results. We provide hereafter a list of differences between [1] and our paper:
>
> [1] considers only quadratic loss. In contrast, in Section 2, we only assume smoothness, PL,  and growth conditions on the “loss” function denoted by $f$ in our work, which all hold  for the special case of quadratic loss.  Note that the growth condition  is a natural assumption to show linear convergence of SGD with constant step-size [2]. In [1], loss function refers to the objective function, which is $h$ in our paper.
>
> In our work, a particular emphasis is on obtaining the best  scaling of the number of the parameters required for convergence of SGD in terms of the sample size. This is done by obtaining bounds on the near-isometry/growth condition constants, as well as choosing proper initialization schemes, which also differ from [1]. Liu et al. [1] find scalings of the width in terms of the number of training examples under a *a different initialization scheme* in Theorem 4. Applying their results to our setting (shallow neural network), one obtains quadratic scaling of the *number of parameters*. In our paper, we show that depending on the batch size, a subquadratic scaling on the number of parameters can be sufficient under *standard initialization*.
>
> They consider a deep network with any number of layers focusing on the minimum width across the network. However, we restrict to a two-layer neural network and exploit the varying width across the network. This helped us improve the scaling on the overall number of parameters.
>
> In terms of proof technique, the authors of [1] used an argument to show that Lipschitz continuity and smoothness of the map lead to “Uniform conditioning” and PL conditions in a neighborhood of initialization. However, we did not use Lipschitz continuity but smoothness of the map to show near-isometry in a ball.
>
> Furthermore, establishing lower bounds on the minimum singular value of the Jacobian and computing the (expected) length of the (S)GD trajectories are common techniques used in the overparameterization literature under various settings ([3-7]). We would like to clarify that we bound the expected length of SGD trajectory under different assumptions compared to [1].
>
> Finally, we highlight that we have clearly cited [Anonymous] whenever we reused results from this work. It is surprising that we were accused of plagiarism based on a work that was not publicly available at the time of submission and was cited in our submission.
>
> [1] Chaoyue Liu, Libin Zhu, and Mikhail Belkin. Loss landscapes and optimization in over- parameterized non-linear systems and neural networks. In Advances in neural information processing systems (NeurIPS), 2020.
>
> [2] Volkan Cevher, Bang Cong Vu. On the linear convergence of the stochastic gradient method with constant step-size.  Optimization Letters volume 13, pages 1177–1187 (2019).
>
> [3] Samet Oymak and Mahdi Soltanolkotabi. Overparameterized nonlinear learning: Gradient descent takes the shortest path? In International Conference on Machine Learning (ICML), 2019.
>
> [4] Samet Oymak and Mahdi Soltanolkotabi. Towards moderate overparameterization: global convergence guarantees for training shallow neural networks. IEEE Journal on Selected Areas in Information Theory, 1:84–105, 2020.
>
> [5] Chirag Gupta, Sivaraman Balakrishnan, and Aaditya Ramdas. Path length bounds for gradient descent and flow. Journal of Machine Learning Research, 22:1-63, 2021.
>
> [6] Simon S. Du, Xiyu Zhai, Barnabas Poczos, and Aarti Singh. Gradient descent provably optimizes over-parameterized neural networks. In International Conference on Learning
> Representations (ICLR), 2019.
>
> [7] Simon Du, Jason Lee, Haochuan Li, Liwei Wang, and Xiyu Zhai. Gradient descent finds global minima of deep neural networks. In International Conference on Machine Learning (ICML), 2019.

---

> > ### Comment · Reviewer_SWMd · 2021-11-25
> > **Reply**
> >
> > Thanks for your response.
> >
> > Part 1: regarding the ethic flag
> >
> > I raised those two points in ethic flags just for a further review. As for the cited anonymous paper, it is for a possible dual submission, not about ‘plagiarism’.
> >
> > Part 2: overlap with prior works and novelty
> >
> > > [1] considers only quadratic loss. In contrast, ….. which all hold for the special case of quadratic loss.
> >
> > I could not agree with this. This submission only considered quadratic loss for neural networks, see section 3. Although other losses are discussed in Section 2, none of them are applied to neural networks.
> >
> > I also noticed that general loss theory based on PL conditions are established in appendices of [1].
> >
> > > the growth condition is a natural assumption to show linear convergence of SGD with constant step-size.
> >
> > I don’t see why growth condition is a natural assumption. Recently it is found to be “unrealistic”, see (A Khaled, 2020).
> >
> > > They consider a deep network with any number of layers focusing on the minimum width across the network. However, we restrict to a two-layer neural network and exploit the varying width across the network. This helped us improve the scaling on the overall number of parameters.
> >
> > The two-layer neural network is a special case of multi-layer networks. So the analysis should be covered by prior works. I don’t see novelty of this submission on this point.
> >
> > > the authors of [1] used an argument to show that Lipschitz continuity and smoothness of the map lead to “Uniform conditioning” and PL conditions in a neighborhood of initialization. However, we did not use Lipschitz continuity but smoothness of the map to show near-isometry in a ball.
> >
> > Lipschitz continuity within a ball should be a weak condition, because the ball radius is finite. I don’t think there is much novelty by removing Lipschitz continuity by some other condition.
> >
> > > establishing lower bounds on the minimum singular value of the Jacobian and computing the (expected) length of the (S)GD trajectories are common techniques used in the overparameterization literature under various settings ([3-7]). We would like to clarify that we bound the expected length of SGD trajectory under different assumptions compared to [1].
> >
> > Although there is some difference in the setting,I don’t see much technical difference between the current submission with prior works, e.g., [1].
> >
> >
> > Reference:
> >
> > A Khaled, 2020: better theory for sgd in the nonconvex world

---

> > > ### Author Response · Authors · 2021-11-28
> > > **Reply to Reviewer SWMd**
> > >
> > > Within the response, the reviewer has acknowledged that there are significant differences between our paper and [Liu et al., 2020] regardless of whether such differences are sufficient in terms of novelty. We would like to ask the reviewer to let us know whether further clarificaitons are needed regarding plagarism concerns. If not, please remove this flag since it is a serious accusation.
> > >
> > > The anonymous paper is already accepted at another venue but was properly cited it since it was not publicly available at the time of submitting this paper.
> > >
> > > Growth condition is shown to be both **necessary** and **sufficient** for convergence of SGD with linear rate [Cevher and Vu, 2019]. Please note that [Khaled and Richtárik, 2020] obtained **sublinear** convergence rate under a more relaxed assumption.
> > >
> > > We emphasize that our paper is the first to show subquadratic scaling is sufficient for linear convergence of SGD under standard initializations and training both layers simultaneously when batch size grows with $m$. Regardless of the general resutls for multi layer NNs, showing such result for the specific case of shallow NNs has not been done to our best knowledge.
> > >
> > > Chaoyue Liu, Libin Zhu, and Mikhail Belkin. Loss landscapes and optimization in over- parameterized non-linear systems and neural networks. In Advances in neural information processing systems (NeurIPS), 2020.
> > >
> > > Volkan Cevher, Bang Cong Vu. On the linear convergence of the stochastic gradient method with constant step-size.  Optimization Letters volume 13, pages 1177–1187 (2019).
> > >
> > > Ahmed Khaled and Peter Richtárik. Better theory for SGD in the nonconvex world. arXiv preprint arXiv:2002.03329 (2020).

---

### Official Review · Reviewer_Vg9p · 2021-11-03

**Correctness:** 3
**Technical Novelty And Significance:** 2
**Empirical Novelty And Significance:** Not applicable
**Recommendation:** 3
**Confidence:** 4

**Main Review:**

While the results of this paper look correct and rigorous, I have a major concern about the significance and novelty of the paper.

First of all, the $O( m^2 / \sqrt{b} )$ condition on overparameterization in fact requires additional assumptions on the data (or data distribution) that is not covered by Assumptions 1-4 in this paper. This result relies on an estimated order of $\sigma_{\mathrm{min}} (X^{*t}) $ that is derived by assuming X is uniform. This is a very strong data distribution assumption. With this additional assumption, it is not fair to compare the condition $\sigma_{\mathrm{min}} (X^{*t}) $ with the other works in Table 1. For example, in order to compare the result with existing works considering the separable data assumption, the authors should derive additional guarantees of $\sigma_{\mathrm{min}} (X^{*t}) $ based on this separable data assumption, as is done in Oymak & Soltanolkotabi (2020).

Moreover, the discussion on Ji & Telgarsky (2020), Chen et al. (2021), and Daniely (2020) is too vague and not convincing. It still seems that these works have already proved better conditions of overparameterization than this paper. Since these works also highlight the improvement of overparameterization conditions, the authors should consider adding them to Table 1 and discuss the differences between this work and existing works in detail.

At last, as given in Table 1 in this paper, the result in this paper is only better than Oymak & Soltanolkotabi (2020) in the sense that (i) this paper considers the training of parameters on both layers, while Oymak & Soltanolkotabi (2020) only considers the training of the first layer (however, in literature, it is widely believed that the difficulty of the analysis lies mainly on the training of the first layer parameters); (ii) this paper gives results for different mini-batch sizes. However, this paper also has a disadvantage compared to Oymak & Soltanolkotabi (2020), as this paper requires a smooth activation function, while Oymak & Soltanolkotabi (2020) works for ReLU activation. Therefore, the result of this paper can be quite incremental compared to Oymak & Soltanolkotabi (2020).


Typos:

1. The sample size is denoted as n in Table 1. However, in the latter part of the paper, the notation of sample size is m.

2. The notation $\nabla h_j(w^i)$ Equation (4) is not consistent with that in the discussion below it $\nabla h_j(\Phi(w^i))$




**Summary Of The Paper:**

This paper focuses on improving the condition of overparameterization for stochastic gradient descent on overparameterized shallow neural networks. The authors first present a general global convergence result for SGD on a finite-sum compositional optimization problem, and then apply this general result to two-layer neural networks with smooth activation functions to obtain a sufficient condition of overparameterization of order $O(m^2 / \sqrt{b})$, where $m$ is the sample size and $b$ is the mini-batch size.

**Summary Of The Review:**

My major concern about this paper is its significance and novelty. Based on this concern, I would like to recommend rejection.

---

> ### Author Response · Authors · 2021-11-19
> **Reply to reviewer Vg9p**
>
> Thank you for your valuable feedback and time spent reading our work. We now go over your main concerns
>
> > **Q1.** the $\mathcal{O}(m^2/\sqrt{b})$ condition [...] requires additional assumptions on the data that is not covered by Assumptions 1-4. This result relies on an estimated order of $\sigma_{\min}(X^{*t})$ that is derived by assuming $X$ is uniform. This is a very strong data distribution assumption.
>
> **A1:** We will clarify the assumptions in our work in Table 1. We note that the analysis based on separable data assumption is different from ours. In general, we cannot find a lower bound on $\sigma_{\min}(X^{*t})$ by merely assuming data separability. Oymak & Soltanolkotabi (2020) analyzed these cases independently of each other.
> ________
> > **Q2.** he discussion on Ji & Telgarsky (2020), Chen et al. (2021), and Daniely (2020) is too vague and not convincing. It still seems that these works have already proved better conditions of overparameterization than this paper. [...] authors should consider adding them to Table 1 and discuss the differences between this work and existing works in detail.
>
> **A2.** Regarding the results in (Ji & Telgarsky 2020, Lemma 4.2) and (Chen et al. 2021, Theorem 3.5): such works analyze the convergence in an ergodic sense (an upper bound on the average of function values). We focus on the value of the objective function at the last iterate, which is a more strict notion of convergence and corresponds more faithfully to the practical training of neural networks. We will include such results in Table 1 and specify the notion of convergence studied.
>
> Daniely (2020) consider binary classification problem and show that SGD on a network with $\tilde O(m)$ number of parameters converges with $O(\frac{m}{\epsilon^2})$ steps.  Because the convergence rate of SGD in that work is sub-linear, it is not directly comparable to our work, where we show a linear rate of convergence for SGD.
> ________
>
> > **Q3.**  the result in this paper is only better than Oymak & Soltanolkotabi (2020) in the sense that (i) this paper considers the training of parameters on both layers, while Oymak & Soltanolkotabi (2020) only considers the training of the first layer (however, in literature, it is widely believed that the difficulty of the analysis lies mainly on the training of the first layer parameters)
>
> **A3.** We think that the most important goal is  to tackle the general case of training all layers, as it corresponds to the algorithms used in practice. There is no formal statement showing that training only the first layer is enough, as far as we know.
> _______
>
> > **Q4.** this paper also has a disadvantage compared to Oymak & Soltanolkotabi (2020), as this paper requires a smooth activation function, while Oymak & Soltanolkotabi (2020) works for ReLU activation. Therefore, the result of this paper can be quite incremental compared to Oymak & Soltanolkotabi (2020).
>
> **A4.** Regarding the assumptions on the activation function and whether our results apply to the popular ReLU activation function, we argue that the differentiability assumption  is not a drawback diminishing the significance of our results because:
> 1) Differentiable activation functions are widely used in practice.
> 2) The differentiability assumption is also used in other overparameterization works (see ref [Liu et al., 2020]).
> 3) The main goal in the literature is to understand when optimization is successful for a neural network of increasing width. Given that the goal is to understand the general behaviour, showing results for differentiable activation functions would translate to our  understanding of ReLU as it can be seen as a limit case if we  consider a sequence of smooth approximations of ReLU.
> 4) Focusing on a particular fixed activation function can be seen as limited, compared to using differentiability assumptions that encompass many different activations used in practice, as well as arbitrary smooth approximations of ReLU.
> 5) Using ReLU makes the whole objective non-smooth which makes it much harder to rigorously talk about the gradient of the function, as we remark in the introduction.
> 6) Finally, works that focus on ReLU activations implicitely assume that SGD never falls in points of non-differentiability. If we include a similar assumption, it is possible to obtain results for ReLU.
>
> Chaoyue Liu, Libin Zhu, and Mikhail Belkin. Loss landscapes and optimization in over- parameterized non-linear systems and neural networks. In Advances in neural information processing systems (NeurIPS), 2020.

---

### Decision · Program_Chairs · 2022-01-20

**Decision:**

Reject

**Comment:**

This paper shows SGD enjoys linear convergence for shallow neural networks under certain assumptions. However, reviewers reach the consensus that this paper lacks technical novelty. The meta reviewer agrees and thus decides to reject the paper.